# Retrosynthesis prediction using an end-to-end graph generative architecture for molecular graph editing

Weihe Zhong[1,2,5], Ziduo Yang[1,5] & Calvin Yu-Chian Chen [1,3,4] ✉

Retrosynthesis planning, the process of identifying a set of available reactions to synthesize the target molecules, remains a major challenge in organic synthesis. Recently, computer-aided synthesis planning has gained renewed interest and various retrosynthesis prediction algorithms based on deep learning have been proposed. However, most existing methods are limited to the applicability and interpretability of model predictions, and further improvement of predictive accuracy to a more practical level is still required. In this work, inspired by the arrow-pushing formalism in chemical reaction mechanisms, we present an end-to-end architecture for retrosynthesis prediction called Graph2Edits. Specifically, Graph2Edits is based on graph neural network to predict the edits of the product graph in an auto-regressive manner, and sequentially generates transformation intermediates and final reactants according to the predicted edits sequence. This strategy combines the two-stage processes of semi-template-based methods into one-pot learning, improving the applicability in some complicated reactions, and also making its predictions more interpretable. Evaluated on the standard benchmark dataset USPTO-50k, our model achieves the state-of-the-art performance for semi-template-based retrosynthesis with a promising 55.1% top-1 accuracy.

Organic synthesis is a central part of several areas of chemistry, including drug discovery, chemical biology, and materials science, which aims to efficiently construct compounds through various organic reactions. Retrosynthesis[1] is a method widely used by organic chemists to design synthetic routes to a target molecule by recursively decomposing it into simpler precursors. Retrosynthesis analysis is a one-to-many problem that is challenging even for experienced chemists due to the huge search space of all possible chemical transformations and the incomplete understanding of the reaction mechanism. Therefore, researchers have been seeking efficient and accurate methods based on the computer-aided synthesis planning (CASP) for decades[2-4]. In recent years, with the rapid development of artificial intelligence (AI) technology and accumulation of chemical data, data-driven methods have sprung up and assisted chemists to save tremendous time and efforts in designing synthetic experiments[5-15].

Existing machine-learning-based retrosynthesis models can be roughly divided into three categories[16,17]: template-based, template-free methods and semi-template-based. A template-based approach is conceptually similar to the process by which organic chemists select a known reaction type to apply to a target molecule. The templates encode the core reactive rules that describe the molecular changes during the reaction and are typically extracted from chemical reaction datasets[18,19]. After a library of reaction templates is constructed, the algorithms match a target molecule with these templates and convert product molecules into reactant molecules by the matched template.

[1]Artificial Intelligence Medical Research Center, School of Intelligent Systems Engineering, Shenzhen Campus of Sun Yat-sen University, Shenzhen 518107, China. [2]School of Biomedical Engineering, Shenzhen Campus of Sun Yat-sen University, Shenzhen 518107, China. [3]Department of Medical Research, China Medical University Hospital, Taichung 40447, Taiwan. [4]Department of Bioinformatics and Medical Engineering, Asia University, Taichung 41354, Taiwan. [5]These authors contributed equally: Weihe Zhong, Ziduo Yang. ✉e-mail: chenyuchian@mail.sysu.edu.cn

Since the selection and application of suitable templates to generate chemically feasible reactants is a more efficient and interpretable way, various works[20–23] have been proposed to use different approaches to prioritize templates. Retrosim[20] ranked the candidate templates based on molecular fingerprint similarity between the target product and the compounds in the corpus. Segler and Waller[21] employed a hybrid neural-symbolic model (Neuralsym) to learn a multi-class classification task for template selection. GLN[22] treated chemistry knowledge of reaction templates as logic rules and learned the conditional joint probability of rules and reactants using graph embeddings. Recently, LocalRetro[23] evaluated the suitable local templates (atom/bond templates) at the predicted reaction centers of a target molecule and considered the nonlocal effects of chemical reactions using global reactivity attention, which achieved the state of the art in the template-based methods. Despite their great potential and interpretability in retrosynthesis prediction, template-based methods have limited coverage due to the inability to predict reactions outside template library and cannot be extended to large-scale template sets because of the expensive computational cost.

By contrast, template-free methods bypass the need to construct an external template database by directly transforming products into potential reactants. Existing works[24–36] in this field recognized that retrosynthesis could be treated as a neural machine translation problem by representing molecules as text, e.g. simplified molecular input line entry system (SMILES) strings[37]. One early example is a sequence-to-sequence (seq2seq) model[24] which converted the SMILES of a product to the SMILES of its reactants by a long short-term memory (LSTM) architecture[38]. Building on this work, subsequent researches achieved better performance by applying a more advanced natural language processing (NLP) model, Transformer[39]. The key drawback of these approaches is that not all generated SMILES strings result in a valid chemical structure. Zheng et al.[26] proposed the SCROP model which added a grammar corrector on the Transformer to attempt to fix the syntax errors of outputs. And to fully exploit the structural information of molecules, Graph2SMILES[30] combined the sequential graph encoder with a Transformer decoder to translate the molecular graphs into the SMILES sequences and showed a comparable accuracy with a template-based baseline model. Compared with the template-based approaches, the template-free methods directly generate the reactant SMILES character-by-character without subgraph matching computation, which have greater generalization potential and a relatively low computational cost. However, linear SMILES representations cannot effectively capture the rich structural information in a molecule, such as the interatomic relationships. And as these models generate SMILES strings by sequentially outputting individual symbols, their predictions are limited in variety and interpretability.

Motivated by chemists' expert experience, semi-template-based approaches[16,40–43] for automating retrosynthesis prediction have recently been developed to address the aforementioned issues. The semi-template-based method is defined as not using a reaction template, nor directly converting the product to the reactant, but predicting the final reactant through the intermediates or synthons generated in multiple steps. Based on the fact that only a small fraction of the molecular structure is modified in a chemical reaction, most existing researches decomposed the retrosynthesis into two steps: first identifying the reaction center using graph neural network (GNN) to form synthons via molecular editing, and then completing the synthons into reactants by either a graph generative model[40], a Transformer[41,42], or a subgraph selection model[16]. These two-stage frameworks enhance the scalability and diversity through simplifying the one-to-many generation problem into multiple one-to-one translation processes, and show promising performance in retrosynthesis prediction task. However, such methods require training two separate modules to complete the transformation, ignoring a strong link between center identification and synthon completion in chemical reactions. Besides, most of them only focus on at

most one atom or bond center, making it challenging to deal with reactions involving multiple centers, which are particularly common in ring formation processes. In contrast, MEGAN[44], an end-to-end framework, modeled the single-step retrosynthesis as a process of applying a sequence of edits to product graph, but the performance was relatively low due to the long edits sequence.

In organic synthesis, it is crucial to understand the reaction mechanism by applying the arrow-pushing approach which simplifies the stepwise electrons shift using sequences of arrows in molecular graphs[45]. As shown in Fig. 1a, a simplified mechanism example in the Mitsunobu reaction: the reagent PPh$_3$ (triphenylphosphine) combines with DEAD (diethyl azodicarboxylate) to generate a phosphonium intermediate that binds to the alcohol oxygen (reactant **2**), activating it as a leaving group, then the nucleophile oxygen anion (**3**) and the phosphonium ion (**4**) to perform nucleophilic substitution to yield the final product (**5**). Based on the approximate reaction mechanism, there have been some machine learning models proposed for forward reaction prediction[44,46–49]. Bradshaw et al.[46] proposed a generative model for reaction mechanism prediction, which formulated the reaction electron paths as a sequence of graph transformations including bond removal and addition. Fooshee et al.[47] also introduce a deep learning approach to predict and rank reaction outcomes through identifying electron sources and sinks. Similarly, GTPN[48] integrated GNN and reinforcement learning (RL) to predict an optimal sequence of operation on atom pairs that transforms the reactants into products. However, most of these methods cannot be directly used for retrosynthetic prediction since no other leaving groups or atoms need to be added in the forward reaction prediction. And it should be mentioned that the semi-template-based MEGAN[44] was the first to model the reaction as an editing sequence for retrosynthesis prediction. Perhaps due to the complex encoder-decoder framework and the add operations at the atomic level, that work made the reactant generation challenging and performed not well in reactions that require attaching the large leaving group, and showed relatively low accuracy on benchmark dataset.

Inspired by the arrow-pushing formalism used in the description of reaction mechanisms mentioned above, we describe retrosynthesis as predicting the reactant graphs by sequentially modifying the product graph based on the simplified mechanisms of reaction transformations. Such a strategy can combine the advantages of both template-based and template-free methods and provide greater interpretability of predictions. It is worth noting that unlike MEGAN model, we simplify the network architecture to effectively learn molecular representations, replace the add-atom actions with attaching substructures to reduce generation steps, and improve the efficiency for generating the reactants.

In this work, we propose a graph-to-edits framework, Graph2Edits, based on simplified reaction mechanisms for retrosynthesis prediction. Specifically, we formulate retrosynthesis as a product-intermediates-reactants reaction reasoning process completed by a series of interconnected graph edits. Our design enables the model to learn the rules of reaction transformation to a certain extent, enhancing the applicability and generalization ability in complicated reactions. Throughout the study, Graph2Edits achieves a top-1 exact match accuracy of 55.1% on the benchmark USPTO-50k dataset, improves the diversity and interpretability of prediction results.

## Results

Following the reasoning logic of chemists, our approach focuses on inferring what local changes occur during the formation of a given product in terms of bond formation or breaking and functional group addition or removal. Therefore, we design an end-to-end architecture (Graph2Edits), based on GNN, to predict a sequence of edits on bonds and atoms of a product molecule. According to the generated edits sequence, the product molecule can be sequentially converted into intermediates and reactants by the RDKit tool[50].

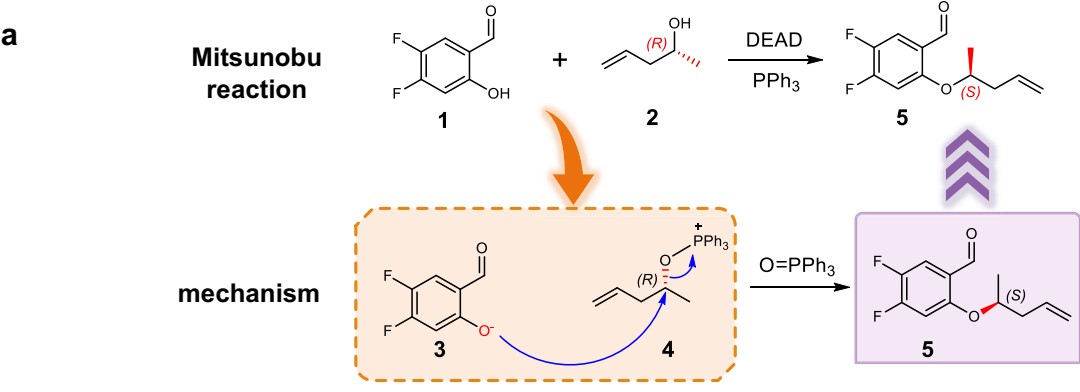

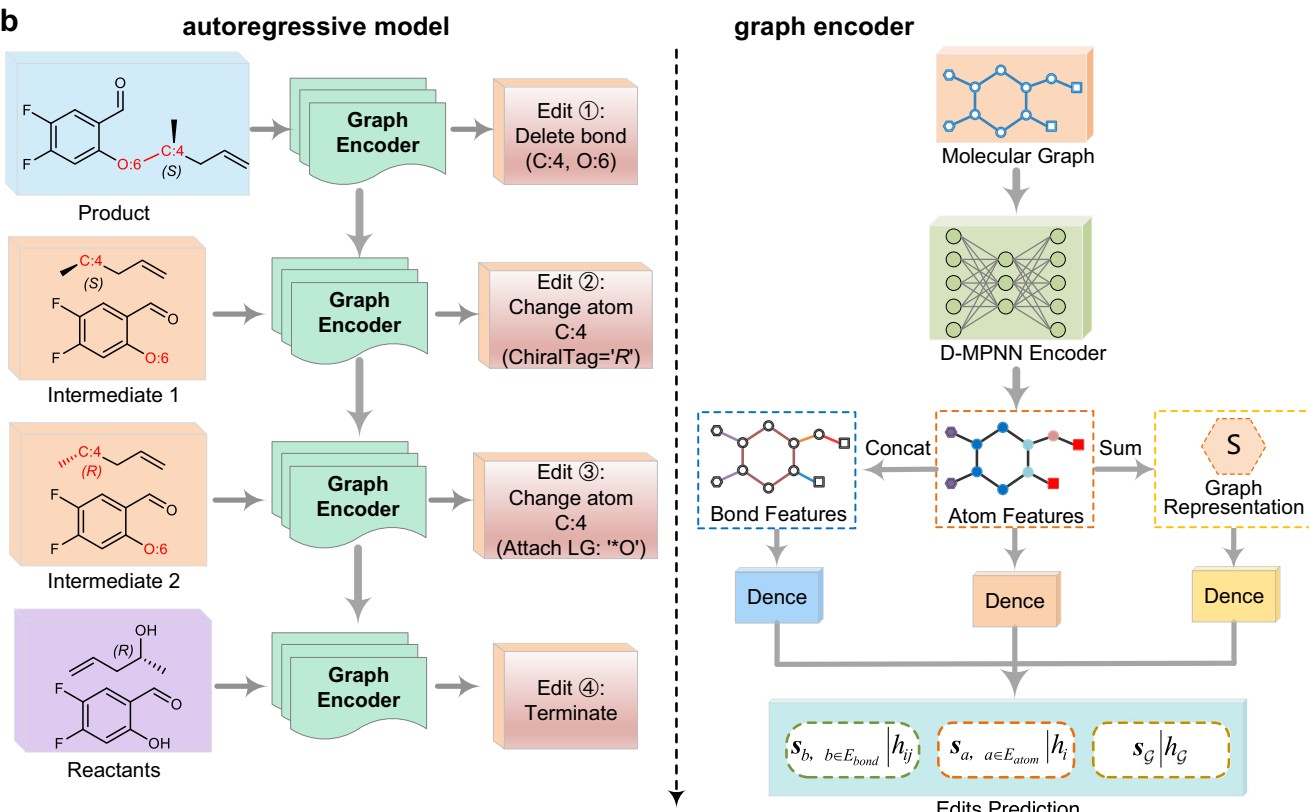

**Fig. 1 | The motivation and overview of Graph2Edits. a** An example of the arrow-pushing formalism in the mechanism reasoning process of Mitsunobu reaction. The steric configuration of the carbon atom in the reaction center is reversed. In the retrosynthesis prediction, DEAD (diethyl azodicarboxylate) and PPh₃ (triphenyl-phosphine) are two reaction reagents that were removed, and we regard the process of the simplified mechanism derivation as sequence of graph edits on the molecular structure. **b** The architecture of Graph2Edits for retrosynthesis prediction. The autoregressive model is for generating edits sequence and the graph encoder is for edits prediction. First, the product molecular graph is encoded by directed message passing neural network (D-MPNN) to generate atom/bond features and graph representations for atom/bond edits and termination symbol predictions. Next, applying the predicted edit on the input graph to obtain intermediate for the following generations. In this example, in the first step, the model predicts to delete the bond between C:4 and O:6, and generates intermediate 1. Subsequently, according to the graph of the intermediates, Graph2Edits predicts the change in chirality and attaches the leaving group '*O' on atom C:4. Finally, the generation process will be complete when Graph2Edits outputs a termination symbol.

## Data preparation and model architecture

We use the publicly available benchmark dataset USPTO-50k[51], containing 50016 reactions with the correct atom-mapping which have been classified into 10 distinct reaction types. We adopt the same split as reported in Coley et al.[20] and divide it into 40k, 5k, 5k reactions for the training, validation, and test sets, respectively. To remove the information leak of USPTO-50k dataset mentioned in the previous studies[16,41], we also canonicalize the product SMILES and re-assign the mapping numbers to the reactant atoms following the method given by Somnath et al.[16]

In order to construct the required output graph, we first derive a set of edits from the USPTO-50k reaction database that can be applied to the input graph. Since the reaction product and reactants are atom-mapped, edits can be automatically extracted by comparing the difference of atoms and bonds between product and reactants. We build the edits vocabulary in the training set, and these edits cover 99.9% of the reactions in the test set, including 6 bond edits, 152 atom edits (7 Change Atom and 145 Attach LG), and a termination symbol:

1. Delete Bond: deletes a bond between two atoms.

2. Change Bond: changes the bond type to single, double, or triple, or changes the stereo configuration of the bond to any, cis or trans.
3. Change Atom: changes the number of hydrogens on an atom to 0, 1, 2, or changes the chiral type of atom to unspecified, *R* or *S*.
4. Attach LG: attaches the functional group called leaving group (LG) to the atom.
5. Terminate: indicates the current molecules are reactants and the generation process terminate.

As in previously reported research[16], few samples in the training set have new bond formations, and we also predict bond edits only for existing chemical bonds rather than for every atomic pair to reduce computational complexity. In general, the prioritization of ground-truth edits for retrosynthesis reactions is consistent with chemical knowledge. Specifically, the atom center reaction shown in Supplementary Fig. 1a is a deprotection reaction and the retrosynthetic transformation is to first reduce the number of hydrogens at the N:1 and followed by attachment of a leaving group (*C( = O)c1ccccc1C(*)=O', the dummy atom * in the leaving group represents the position of attaching). Supplementary Fig. 1b shows an example of bond center reactions, and in this retro-reaction, a C – C bond is removed and connected by a Br and a dimethylamino group respectively. For multiple centers reactions in Supplementary Fig. 1c, the edits sequence is organized by breaking the bond, followed by changing the property of the atom or bond, and finally attaching the leaving group. More details about the graph edits could be found in Section Methods, Supplementary Data 1, and Supplementary Fig. 2.

Additionally, we also use the original USPTO-full dataset from the entire USPTO (1976-Sep2016) to verify the scalability of our model. We use exactly the same splits as Dai et al.[22], which contain approximately 800k/100k/100k training/validation/test reactions, and repeat procedures given in the above USPTO-50k dataset processing.

We employ the directed message passing neural network (D-MPNN)[52], a variant of the generic message passing neural network (MPNN)[53], to obtain the atom representations and then utilize the integrated local atom/bond and global graph features to predict atom/bond edits and a termination, respectively. The overall inference process of Graph2Edits is shown in Fig. 1b.

## Performance evaluation

We adopt the top-k exact match accuracy as the metric to evaluate the retrosynthesis performance. The exact match accuracy is computed by comparing the canonical SMILES of predicted reactants to the ground truth in the dataset. We additionally adopt the round-trip[31] and MaxFrag[32] accuracy to evaluate the performance of our model. The round-trip accuracy is calculated by comparing the ground-truth product with the product predicted by a forward reaction prediction model using the predicted reactants, and is to evaluate the correctness of the predictions generated by the retrosynthetic model as there might be multiple different reactants can be used to synthesize the same product. We here use the pretrained forward-synthesis prediction model Molecular Transformer (MT)[54] to evaluate the round-trip accuracy. The MaxFrag accuracy, inspired by classical retrosynthesis, is to calculate the exact match of only the largest fragment to overcome the prediction limitation due to the existence of unclear reagent reactions in the dataset. Considering the changes of stereochemistry in the reactions, we retain the chirality and cis-trans isomer information in the molecule for comparison. And for evaluating the overall performance, we compare the prediction results of Graph2Edits with several template-based, template-free, and semi-template-based methods, including current state-of-the-art models. Semi-template-based G2G[40], RetroXpert[41], RetroPrime[42], MEGAN[44] and GraphRetro[16] are primary baselines as their design ideas use a similar two- or multi-step generation and achieve excellent performance. To show the broad superiority of model, we also take the template-based Retrosim[20], Neuralsym[21], GLN[22], LocalRetro[23] and template-free SCROP[26], Augmented Transformer[32], GTA[29], Graph2SMILES[30] and Dual-TF[33] as strong baseline models for comparison.

The results of top-k exact match accuracy on the USPTO-50k benchmark are shown in Table 1. To avoid over-tuning and giving overly optimistic results, we only report the test results for models

**Table 1 | Top-k exact match accuracy of the proposed Graph2Edits and baselines on USPTO–50k dataset**

| Model | Top-k accuracy (%) | | | | | | | | | |
|---|---|---|---|---|---|---|---|---|---|---|
| | Reaction class unknown | | | | | Reaction class known | | | | |
| | k = 1 | 3 | 5 | 10 | 50 | 1 | 3 | 5 | 10 | 50 |
| Template-Based Methods | | | | | | | | | | |
| Retrosim | 37.3 | 54.7 | 63.3 | 74.1 | 85.3 | 52.9 | 73.8 | 81.2 | 88.1 | 92.9 |
| Neuralsym | 44.4 | 65.3 | 72.4 | 78.9 | 83.1 | 55.3 | 76.0 | 81.4 | 85.1 | 86.9 |
| GLN | 52.5 | 69.0 | 75.6 | 83.7 | 92.4 | 64.2 | 79.1 | 85.2 | 90.0 | 93.2 |
| LocalRetro | 53.4 | 77.5 | 85.9 | 92.4 | 97.7 | 63.9 | 86.8 | 92.4 | 96.3 | 97.9 |
| Template-Free Methods | | | | | | | | | | |
| SCROP | 43.7 | 60.0 | 65.2 | 68.7 | – | 59.0 | 74.8 | 78.1 | 81.1 | – |
| Aug. Transformer | 53.2 | – | 80.5 | 85.2 | – | – | – | – | – | – |
| GTA | 51.1 | 67.6 | 74.8 | 81.6 | – | – | – | – | – | – |
| Graph2SMILES | 52.9 | 66.5 | 70.0 | 72.9 | – | – | – | – | – | – |
| Dual-TF | 53.6 | 70.7 | 74.6 | 77.0 | – | 65.7 | 81.9 | 84.7 | 85.9 | – |
| Semi-Template-Based Methods | | | | | | | | | | |
| G2G | 48.9 | 67.6 | 72.5 | 75.5 | – | 61.0 | 81.3 | 86.0 | 88.7 | – |
| RetroXpert[a] | 50.4 | 61.1 | 62.3 | 63.4 | 64.0 | 62.1 | 75.8 | 78.5 | 80.9 | 83.5 |
| RetroPrime | 51.4 | 70.8 | 74.0 | 76.1 | – | 64.8 | 81.6 | 85.0 | 86.9 | – |
| MEGAN | 48.1 | 70.7 | 78.4 | 86.1 | 93.2 | 60.7 | 82.0 | 87.5 | 91.6 | 95.3 |
| GraphRetro | 53.7 | 68.3 | 72.2 | 75.5 | – | 63.9 | 81.5 | 85.2 | 88.1 | – |
| Graph2Edits (MPNN) | 52.7 | 77.2 | 85.3 | 91.0 | 97.1 | 65.7 | 87.3 | 92.0 | 95.3 | 97.8 |
| Graph2Edits (D-MPNN) | 55.1 | 77.3 | 83.4 | 89.4 | 92.7 | 67.1 | 87.5 | 91.5 | 93.8 | 94.6 |

Graph2Edits (MPNN) and Graph2Edits (D-MPNN) use the message passing neural network (MPNN) and the directed message passing neural network (D-MPNN) as graph encoder, respectively.
[a]The results are taken from https://github.com/uta-smile/RetroXpert.

with the highest top-1 accuracy during validation. When the reaction class is unknown, our method achieves a 55.1% top-1 accuracy which outperforms all the baseline models, and for larger k (k = 3, 5, 10, 50), Graph2Edits also beats prior models by a large margin except for the LocalRetro model. For a more precise comparison, Graph2Edits reaches the state-of-the-art performance for semi-template-based methods and is more accurate than GraphRetro and MEGAN model by a margin of 1.4% and 7.0% respectively in top-1 accuracy. With the reaction class given, Graph2Edits outperforms all baselines in all metrics with the exception of top-5, -10 and -50 accuracy in template-based LocalRetro. As shown in the table, our method is ultimately superior to the other semi-template-based models and exceeds the GraphRetro by 3.2% and MEGAN by 6.4% with a 67.1% top-1 accuracy. In addition, although the higher accuracies at higher k have been achieved in MPNN-based models as the redundancy in node messages passing[52,55] may help to improve the probability of predicting the ground-truth leaving group on the reaction centers, using D-MPNN encoder has a clear advantage over conventional MPNN, yielding improvements of 1.4 and 2.4 points on top-1 accuracy with and without giving reaction class, respectively. It is worth noting that in the semi-template-based methods, Graph2Edits not only improves the performance on top-1 accuracy, but also has more advantages on top-k (k > 1) accuracies, and it can be observed that the top-3 accuracy is higher than top-10 accuracies of GraphRetro and G2G model without reaction type given. We deduce that the advantages of Graph2Edits are largely derived from strengthening the correlation between the generation steps and efficiently expanding the search of the diverse reaction space by sequentially editing and attaching substructure on atoms and bonds.

The results of round-trip and MaxFrag accuracy of our model tested on USPTO-50k are shown in Table 2. The top-1 round-trip accuracy of our model reaches nearly 86%, which is comparable to GraphRetro and outperforms MEGAN by a large margin. Additionally, perhaps due to the detailed difference of the calculation methods, the round-trip accuracies of the LocalRetro[23] for USPTO-50k seem to be higher than our results. As there is no related code for calculating the round-trip accuracy in LocalRetro GitHub, in order to make a fair comparison in the semi-template-based methods, we calculate the round-trip accuracies based on the trained models provided by MEGAN and GraphRetro, and provide the LocalRetro's round-trip

accuracy results as a reference. Graph2Edits also beats prior semi-template-based models on top-3, -5, -10, and -50 predictions. For MaxFrag accuracy, Graph2Edits outperforms all baselines by a large margin and achieves 59.2% accuracy at top-1 predictions.

We also compare the performance of Graph2Edits on the larger USPTO-full dataset with other baselines for retrosynthesis prediction. The results are presented in Supplementary Table 2. Although the USPTO-full is much noisier than the clean USPTO-50k, our method still has competitive performance with a top-1 accuracy of 44.0%, on par with the semi-template-based method RetroPrime and outperforming MEGAN by a large margin. In addition, on larger k (k > 1), especially top-10 accuracy, Graph2Edits significantly outperforms all other methods except Aug.Transformer, showing similar superiority to the performance on the USPTO-50k dataset.

## Analysis of correct and incorrect predictions
To more comprehensively understand the model performance, we conduct an error analysis of predictions on the USPTO-50k test set. First, 100 random reactions where the results predicted by Graph2Edits differ from the ground-truth reactants are analyzed by professional organic chemists. The assessment gives 85% of the reactions in which the predicted reactants are feasible and considered correct by the chemists, and interestingly, this result is close to the top-1 round-trip accuracy described previously. We here present 30 random examples in Supplementary Table 3 and display that the proposed reactants by Graph2Edits are difficult to distinguish from the ground-truth reactants in terms of reaction feasibility. To further analysis of the incorrect predictions, we then show some reaction samples in Fig. 2 and find that the most common reason for error predictions is ignoring the influence by other functional group in the molecular structure. The prediction by our model in Fig. 2a may fail due to the low reactivity of secondary amine and the steric hindrance of benzyl group. In Fig. 2b, a more nucleophilic aromatic amine group can lead to a completely different product. And also, Graph2Edits sometimes fails to detect multiple reaction sites, possibly resulting in low yield and some by-products (Fig. 2c). These results indicate that there is still significant scope for improvement in the performance of retrosynthesis prediction, such as introducing more chemically meaningful modules to capture the molecular structure information and identify the reactivity of different reaction sites.

In addition, we visualize the top-10 predictions which are different from the ground truth reactants for two cases in Supplementary Fig. 3. We can observe that the common feature of these two products is to have multiple possible reaction centers, and thus can be yielded through a variety of different reaction types. In fact, all top-10 predicted reactants are feasible and can be synthesized by standard methods, although the reaction yields may vary. In Supplementary Fig. 3a, our model provides the options of replacing 'I' with 'Cl' and 'Br' on top-3 and top-7 prediction and amide condensations on top-1 and top-5 prediction. And in Supplementary Fig. 3b, it is worth emphasizing that the ground-truth reactants in USPTO-50k test set is probably wrong, as it is unlikely to introduce stereochemistry far from the reaction center. And Graph2Edits successfully proposes reactions all start from chiral substrates and the top-2 prediction is perfectly fine. Furthermore, we conduct a more in-depth performance comparison with the baseline model MEGAN and show a comparison of the reaction examples presented by MEGAN in Supplementary Fig. 4. We observe that the top-1 prediction for the first three reactions by our model are feasible and completely consistent with the ground-truth reactants. And although the top-1 prediction for the last reaction is similar to those by MEGAN, the subsequent top-2 prediction by our method provides a decent alternative. Moreover, we also evaluate the invalid rates generated by Graph2Edits and the results can be seen in Supplementary Notes and Supplementary Table 4.

**Table 2 | Top-k Round-Trip and MaxFrag accuracy of the proposed Graph2Edits and baselines on USPTO-50k dataset**

| Category | Model | Top-k (%) | | | | |
|---|---|---|---|---|---|---|
| | | k = 1 | 3 | 5 | 10 | 50 |
| Round-Trip accuracy | LocalRetro[a] | 89.5 | 97.9 | 99.2 | – | – |
| | MEGAN[b] | 82.0 | 89.9 | 91.7 | 94.0 | 96.4 |
| | GraphRetro[b] | 86.0 | 89.9 | 90.7 | 91.4 | 91.6 |
| | Graph2Edits (MPNN) | 84.9 | 93.6 | 95.7 | 96.9 | 98.9 |
| | Graph2Edits (D-MPNN) | 85.9 | 93.5 | 95.1 | 96.4 | 97.3 |
| MaxFrag accuracy | Aug.Transformer | 58.5 | 73.0 | 85.4 | 90.0 | – |
| | LocalRetro | 57.8 | 82.1 | 89.7 | 95.0 | 98.4 |
| | MEGAN | 54.2 | 75.7 | 83.1 | 89.2 | 95.1 |
| | Graph2Edits (MPNN) | 56.8 | 80.3 | 87.5 | 92.8 | 95.4 |
| | Graph2Edits (D-MPNN) | 59.2 | 80.1 | 86.1 | 91.3 | 93.1 |

Graph2Edits (MPNN) and Graph2Edits (D-MPNN) use the message passing neural network (MPNN) and the directed message passing neural network (D-MPNN) as graph encoder, respectively.
[a]The results are taken from the LocalRetro paper.
[b]The results are implemented based on the available trained models in the open-source code.

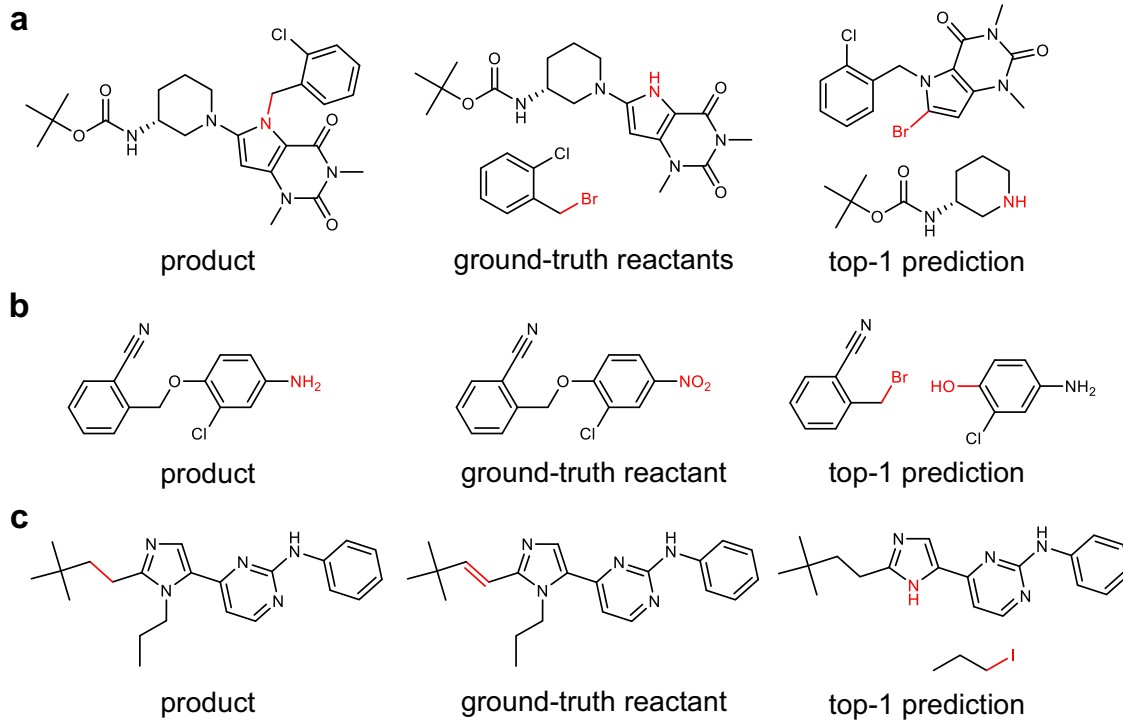

**Fig. 2 | Examples of top-1 prediction by Graph2Edits for different errors. a** The low reactivity of secondary amine and the steric hindrance of benzyl group, **b** Ignoring a more nucleophilic aromatic amine group, and (**c**) fail to detect multiple reaction sites.

## Effect of edits sequence length and stereochemistry

We further conduct more in-depth studies to exhibit the superior performance and generalization of our proposed Graph2Edits on retrosynthetic prediction. Specifically, we investigate the performance effect of some complex reactions in the USPTO-50k, including reactions with long edits sequence length and stereochemistry.

According to the edits sequence length of reactions preprocessed on the test set, we present the distribution of data and top-10 accuracy in Fig. 3. Similar to the distribution of reaction types reported previously[22], the distribution of reactions with various edits length is highly unbalanced. As is shown in Fig. 3a, most reactions have an editing length of 2, 3, or 4, with 207 (4.1%), 3938 (79.7%), 702 (14%) pieces of data, respectively. And the reactions with edits sequence length 5, 6, 7 or longer account for a small proportion, which have 93 (1.9%), 30 (0.6%), 7 (0.1%) and 27 (0.5%) cases respectively. From Fig. 3b we can see the performance of our model does not decrease significantly with the increasing edits length, especially for the situations with small amounts of data. For reactions with 8 or longer edits length, the top-10 accuracy still achieves 81.5%, indicating that the continuous generation of Graph2Edits remains relatively robust even in the complicated reactions. These results demonstrate that our performance is not obtained by overfitting to one particular category of reactions.

As revealed by MTExplainer[56], scaffold bias in the USPTO dataset, where similar molecules appear in both the training and the test set and undergo similar transformations, makes the models achieve high accuracy and does not reflect the true generalization performance of the models. To remove the structural bias and further investigate the performance on diverse reaction products, we re-split the USPTO-50k dataset via the Tanimoto similarities[57] of the reaction products to train the retrosynthetic prediction models. Following the Tanimoto-based splitting given by MTExplainer, the initial USPTO-50k dataset is randomly split 85%:15%, and for the Tanimoto similarity threshold σ = 0.6 and σ = 0.4, the ratios after Tanimoto splitting are 88.3%:11.7% and 95%:5%, respectively. We then train our Graph2Edits along with the other semi-template-based models (MEGAN and GraphRetro) on these

two datasets. Table 3 shows that although the performance of both our Graph2Edits and the baselines decrease upon the new train/validation/test split datasets, our model still outperform MEGAN and GraphRetro by a large margin. These results show that our model could also achieve relatively good generalization performance on the structurally diverse test set.

Stereochemistry plays a significant role in organic chemistry and is also important in drug discovery. It is challenging to predict the change of stereochemistry in the reaction. We count 157 reactions containing the change in stereochemistry in USPTO-50k test set and check them one by one. We found that more than half (51.6%) of ground-truth reactions gave wrong stereochemical information, which is consistent with the noisy stereochemical data reported by Schwaller et al.[31], and in 82.2% of the reactions, the top-1 prediction proposed by Graph2Edits was considered correct by experienced organic chemists. We show the 30 random reactions in Supplementary Table 5, and display that our method performed well on the chiral substrate-induced asymmetric reactions (examples 4, 8, 20), chiral auxiliary-induced asymmetric reaction (example 26), asymmetric hydrogenations (examples 24, 30). Although this stereochemical data set is too limited to claim the performance on stereochemistry, these assessments offer strong evidence that our model has an advantage in predicting stereoselective reactions and can learn some rules of stereochemistry changes.

## Analysis of model reasoning process

To better understand the reasoning process of Graph2Edits, we randomly select 3 reactions with different reaction types from the test set of USPTO-50k and visualize the generation predictions in Fig. 4. The first example is the Suzuki cross-coupling reaction, which describes the formation of a carbon–carbon bond between a halocarbon and a borate ester. Our model predicts a C-C bond break with a high probability of 0.97 and then the top-1 and 3 predictions are to attach the bromine and borate ester in a different order for producing the ground truth. It is worth noting that the top-2 result provides a solution for a

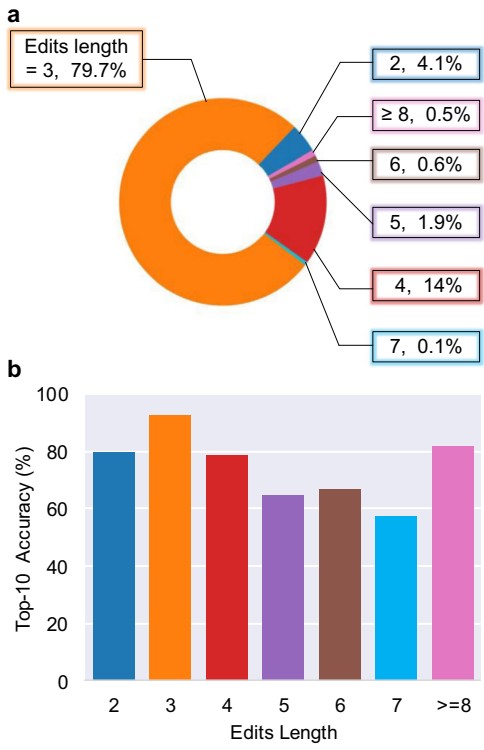

**Fig. 3 | The performance effect of edits sequence length. a** The distribution of different edits sequence length reactions in the test set, (**b**) Top-10 accuracy per each edits length. type. The source data are obtained from the USPTO-50k test set and well-trained model, and can be found in Supplementary Data 2.

**Table 3 | Evaluation of single-step retrosynthetic models on different train-test splits of USPTO-50k dataset**

| Data split | Model | Top-k accuracy (%) | | | |
|---|---|---|---|---|---|
| | | k = 1 | 3 | 5 | 10 |
| Original random split | MEGAN | 48.1 | 70.7 | 78.4 | 86.1 |
| | GraphRetro | 53.7 | 68.3 | 72.2 | 75.5 |
| | Graph2Edits (D-MPNN) | 55.1 | 77.3 | 83.4 | 89.4 |
| Tanimoto similarity <0.6 | MEGAN[a] | 47.0 | 69.2 | 76.2 | 83.6 |
| | GraphRetro[a] | 49.1 | 63.2 | 66.9 | 69.1 |
| | Graph2Edits (D-MPNN) | 52.0 | 75.6 | 83.2 | 89.4 |
| Tanimoto similarity <0.4 | MEGAN[a] | 45.4 | 68.4 | 76.9 | 84.6 |
| | GraphRetro[a] | 44.2 | 56.2 | 58.7 | 59.6 |
| | Graph2Edits (D-MPNN) | 47.5 | 71.7 | 80.1 | 88.0 |

Graph2Edits (D-MPNN) uses the directed message passing neural network (D-MPNN) as graph encoder.
[a]Denotes that the result is implemented by the open-source code with well-tuned hyperparameters.

(top-1 and top-2), which matches the ground-truth reaction. In addition, our method further offers the options of the hydroxyl protection and the aromatic coupling reaction. In the last example, for the reaction of the amide dehydration to form the cyano group, our approach generates the ground-truth reactants in top-1 prediction, and can also provide the heterocycle formation, amino protection and double bond reduction with multiple distinct substrates.

To quantitatively analyze the diversity of predictive results, we investigate the molecular similarities among them. For each product, the similarity is quantified by the mean Tanimoto similarity between the predicted reactants and other top-10 predictions, based on the concatenated ECFP4 fingerprints, and the lower similarity indicates the higher diversity of predicted results. We also use the K-means clustering algorithm to cluster the products according to the similarity of predicted reactants, similar to that used by Chen et al.[43]. As shown in Fig. 5, the first four clusters (dark red to orange) have lower prediction similarities (0.22, 0.36, 0.44, and 0.50), which can be regarded as high-diversity clusters, accounting for about 30% of the test set. The average similarity on middle three clusters (light orange and light blue) is 0.55, 0.60, and 0.65, respectively, and thus can be referred to as medium-diversity clusters, accounting for nearly 54% in test set. And the last three clusters (dark blue), considered as low-diversity clusters, have a small proportion and relatively higher prediction similarities (0.71, 0.80, and 0.98). These results clearly show that Graph2Edits can predict diverse results.

## Graph embedding visualization

To further evaluate the interpretability of the model, we explore the performance of the molecular embedding representation learned by Graph2Edits at each edit step. Specifically, we randomly select 50 reactions with edits length 2, 3, 4, and 5, respectively, and together with all reactions with edits length greater than or equal to 6, a total of 263 reactions from the test set. The product graphs of these reactions are fed into Graph2Edits for generating the high-dimensional features with a 256-dimensional embedding at each edit step. The high-dimensional vector, similar to the fingerprint vector representation of a molecule, is reduced to the 2D embedding space by t-distributed neighbor embedding (T-SNE)[58]. Figure 6 shows the distribution visualization of molecular embeddings at each edit step, and the a−d represents the test results of these reactions on training epochs 5, 25, 50, and 123 (best validate accuracy epoch).

At the beginning of model training, the initialization parameters are roughly optimized for multi-step edits generation and the

boronic acid substrate instead of a boronic ester. The second is Paal-Knorr reaction for the pyrrole synthesis. Our retrosynthesis prediction is first to delete the two bonds of the pyrrole ring, followed by changing the type of bond from double bond to single bond, and finally attach two double bond oxygen groups to generate the reactants. Although this generation process goes through 7 steps, each step generates the correct edit with high probability, which further demonstrates the robustness of our model to continuous inference edits. Another challenging example is the Mitsunobu reaction for synthesis of ether accompanied by the reversal of chiral configuration. Graph2Edits successfully predicts a change in chirality after ether bond breaking and infers candidates with an overall high score. More examples of predictions can be found in Supplementary Fig. 5.

## Diversity on predicted reactants

Evaluation of the diversity of the predicted reactions is crucial, as it is related to whether the predictions of our method can cover a broad range of chemical reactions in multi-step retrosynthetic route planning. Benefiting from our design strategy, Graph2Edits can continuously generate graph edits in an autoregressive manner, and output multiple different reaction centers and leaving groups in beam search, thus enabling the ability to predict reactants with different scaffolds and structures. To analyze the diversity of predicted results, we first present three examples of diverse reactants predicted by Graph2Edits in Supplementary Fig. 6. The first example is 1, 3-dipolar cycloaddition reaction. Our model predicts four different reaction centers, including a nitrogen atom in triazole (top-1, 3, 4, 6, and 9), the whole triazole ring (top-2 and top-5) and two carbon-carbon bonds between aromatic rings (top-7 and top-8). And among these results, three reaction types (the amino protection with different protective groups, 1, 3-dipolar cycloaddition and the Suzuki cross-coupling reaction) are predicted to yield the product. In the second example, Graph2Edits suggests a reduction of the ethyl ester or methyl ester

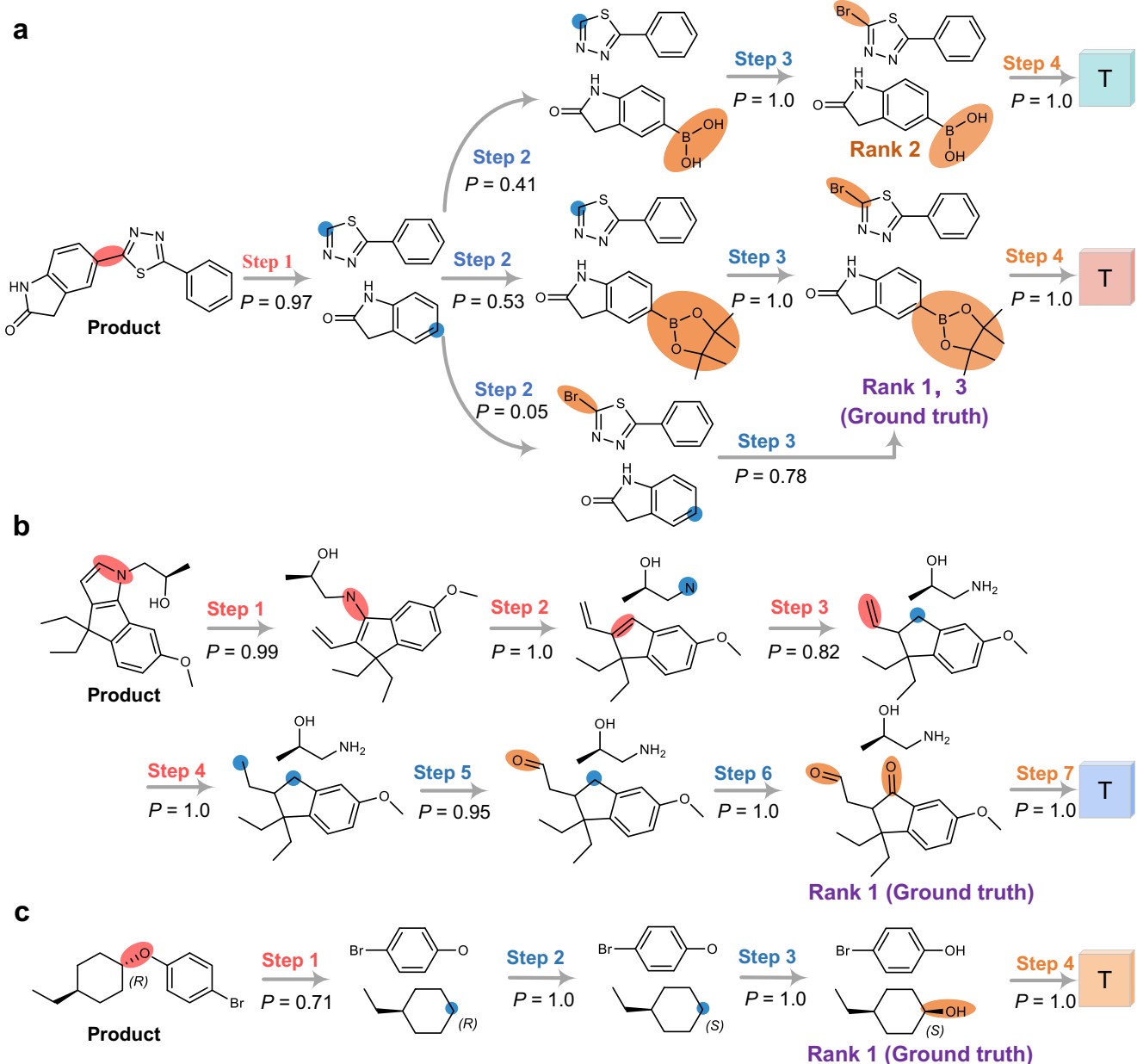

**Fig. 4 | Retrosynthesis reasoning predictions by our model. a** Suzuki coupling reaction, (**b**) Paal-Knorr reaction, and (**c**) Mitsunobu reaction. The '*P*' is the probability of model prediction, and the 'T' represents a termination symbol. The red, blue, and orange steps represent reaction center identification, leaving group attaching, and generating termination, respectively.

intermediates molecular representations over edit steps are still in a mixed state in 2D mapping space at epoch 5 (Fig. 6a). Notably, the generation process of reactions with long edit steps is likely to terminate in small editing steps, indicating that the model has not yet learned the transformation law of the complex reactions. After 20 epochs training (Fig. 6b), the mixing degree of red dots and blue dots weakens and displays aggregation phenomenon to some extent, especially for the molecular representations in the first edit step (red dots). Subsequently, it has been clearly observed in Fig. 6c that the model can better distinguish the molecular vectors in the first and second edit step (red and blue dots), and shows that the Graph2Edits iterations are optimizing in the right direction and learn the underlying rules of reaction. Finally, the model has reached the best performance on retrosynthesis prediction task at epoch 123 (Fig. 6d), and the molecular representations in the first edit step are gathered in the upper left corner of the space. As the editing step lengthens, the molecular representations

move to the lower right of the space, and further illustrate why the model can also perform well in complex reactions with long edit steps. These results suggest that our model can perceive the molecular characteristics on different edit steps for retrosynthesis prediction.

**Multistep retrosynthesis prediction**

To verify the practical use in synthesis planning, we also extend our one-step model trained on the USPTO-50k dataset to full pathway design by sequentially performing retrosynthetic predictions. We choose 3 target compounds as examples, all of which have significant medicinal importance, including the oral SARS-CoV-2 M^pro inhibitor Nirmatrelvir for treatment of COVID-19[59], the third-generation EGFR inhibitor Osimertinib for treatment of non-small cell lung carcinoma[60] and the Lenalidomide for treatment of multiple myeloma[61]. Note that none of these input structures (products and intermediates) in the three examples appears as a product in our training set. As shown in

Fig. 7, our method successfully reproduces the complete synthetic pathway for these compounds.

The first example for Nirmatrelvir has been reported in the literature by Pfizer[62] (Fig. 7a). Although the synthetic pathway consists of six reaction steps, our method succeeds at the rank-1 prediction for all steps except the third one predicted at rank-6, which directly demonstrates the superiority of our method. The first and second steps, which are the core reactions, can be easily reproduced by our model as dehydration of the amide to form the cyano group, followed by a condensation reaction to yield the key intermediate (**6**). The subsequent step is an amine ester exchange reaction, preceded by the common deprotection and ester hydrolysis, and the final step involves the amide formation, which exactly matches the published synthesis. The second example is the retrosynthetic pathway planning of Osi-

mertinib, as depicted in Fig. 7b. Finlay et al.[63] proposed a five-step reaction pathway for this drug, which is derived from readily available starting materials. Our model first suggests an acylation reaction with acryloyl chloride (**14**) and then correctly predicts a reduction of the nitro group with rank-1. In the next two steps, sequential nucleophilic aromatic substitution reactions ($S_N$Ar) are predicted to introduce amino side chain and nitroaniline. And the final step, unlike the Friedel-Crafts arylation reported in the literature, our model suggests a Suzuki cross-coupling reaction to produce 3-pyrazinyl indole (**20**). In the third example, the retrosynthesis pathway planning for Lenalidomide has also been demonstrated by Retrosim[20] and LocalRetro[23] models, and our model can perfectly recover the route suggested by the Retrosim method. The first and third steps are suggested as the nitro reduction and the bromination with N-bromosuccinimide (**26**), which are also consistent with published literature pathway[64]. And in the second step, our model predicts a formation of the five-membered ring with the acid chloride (**25**), rather than the methyl ester, which is feasible in synthetic chemistry. These results clearly show that our approach can generate nearly identical retrosynthetic pathways as those in the literature, mostly within the rank-2 predictions, and further demonstrate the great potential of our model for practical multistep retrosynthesis.

## Discussion

In this study, we developed an end-to-end semi-template-based retrosynthesis prediction model, Graph2Edits, which predicts a possible sequence of edits from the product graph and sequentially generates the intermediates and reactants. In contrast to previous template-based methods that limit predictions to template sets and template-free models that fail to capture the rich structural information in molecular graph, Graph2Edits is a graph-based model that treats one-step retrosynthesis as applying a sequence of graph edits to product graph and generates reactant molecules just as chemists think about how a reaction happened. Comprehensive evaluations on the benchmark dataset USPTO-50k demonstrate that our method achieves a

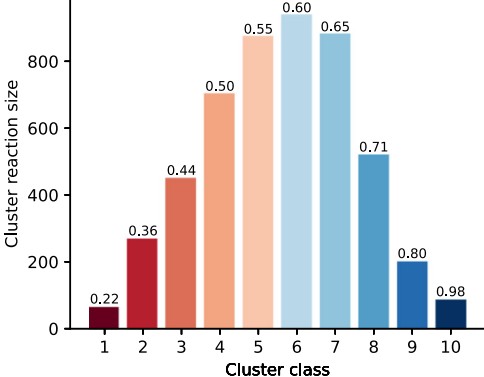

**Fig. 5 | The cluster results on USPTO-50k test set based on predicted reactants similarities.** The numbers above the bars represent the average similarity of the predicted reactants, and the lower the similarity, the higher the diversity. The source data are obtained from the USPTO-50k test set and well-trained model, and can be found in Supplementary Data 3.

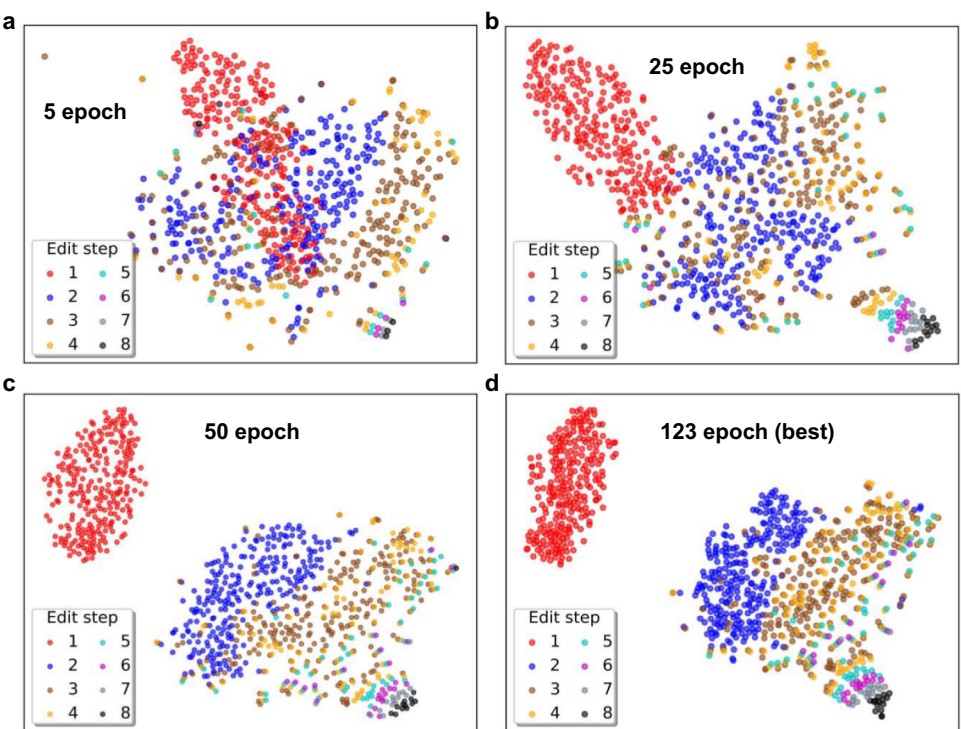

**Fig. 6 | Visualizations of molecular embeddings generated by Graph2Edits at each edit step during the learning process.** The test results on training epochs (**a**) 5, (**b**) 25, (**c**) 50, and (**d**) 123 (best validate accuracy epoch). The 256-dimensional

graph embeddings at each edit step in the training process are reduced to a 2D embedding space by using T-SNE.

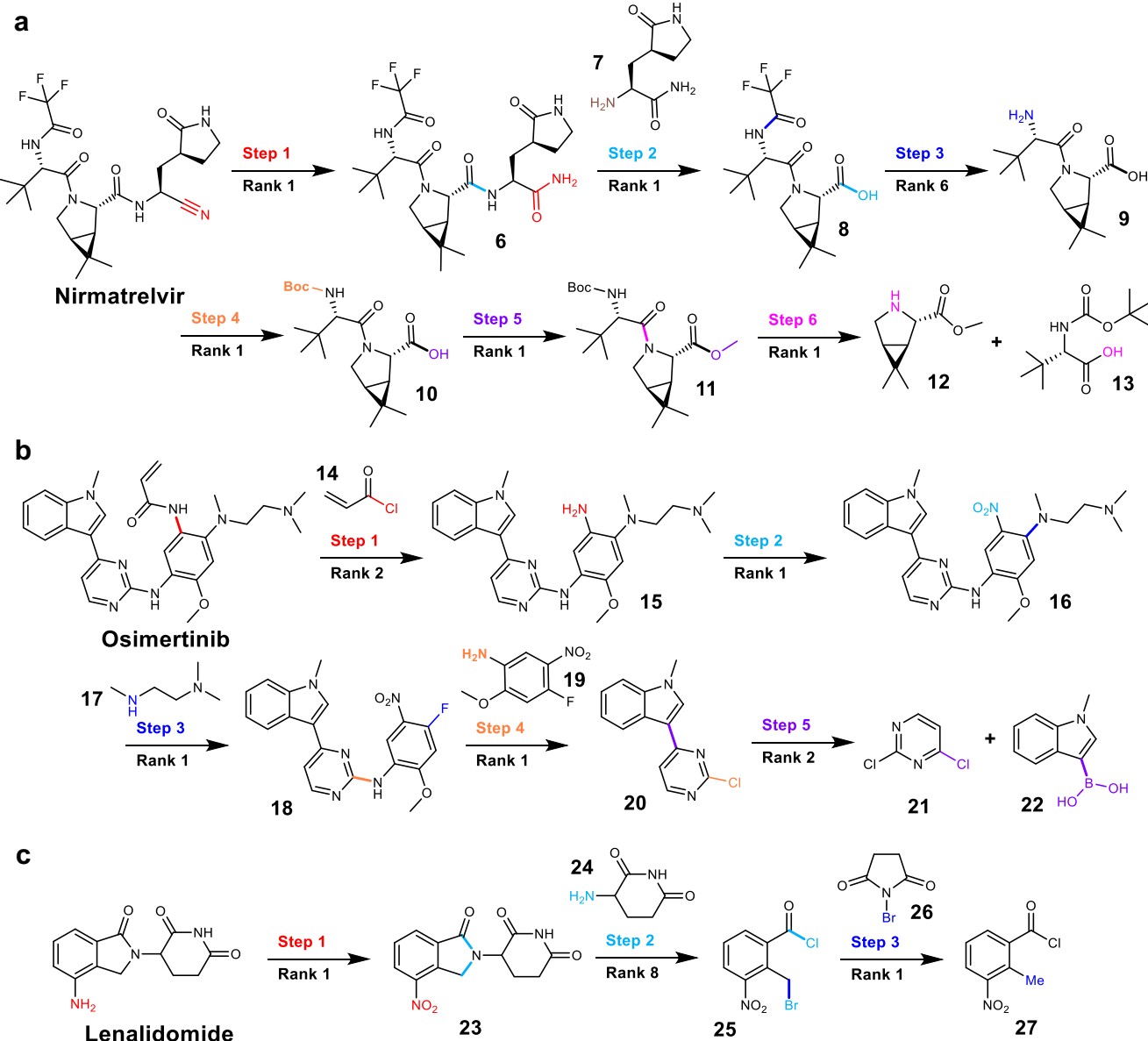

**Fig. 7 | Multistep retrosynthesis predictions by Graph2Edits. a** The oral SARS-CoV-2 M^pro inhibitor Nirmatrelvir, (**b**) The third-generation EGFR inhibitor Osimertinib, and (**c**) Lenalidomide. The reaction center in the atom and bond transformations are highlighted in different colors at different reaction steps.

promising 55.1% top-1 exact match accuracy and shows comparable or improved performance compared to the other state-of-the-art models. In the large and noisy USPTO-full dataset, Graph2Edits also achieves the top-1 accuracy of 44.0%, which is significantly higher than the baseline MEGAN, and is close to the state-of-the-art models. These encouraging results display that our model has excellent generalization and robustness. Crucially, since the multi-step generation predicts arbitrary length edits, the model can more efficiently search the latent space of the plausible reactions and improve the diversity of prediction results. Extensive experiments verified the superiority of the proposed method in some complicated reactions. In particular, detailed analyses of model predictions including molecular representations suggest that this strategy can enhance the rationality and interpretability of retrosynthetic models. Our main contributions are as follows:

(a)  We propose Graph2Edits, an end-to-end architecture that generates arbitrary length graph edits in an auto-regressive way, to combine the center identification and synthon completion

processes into a one-pot learning and improve the applicability in reactions containing multiple reaction centers.

(b)  We introduce a D-MPNN which encodes the local atom/bond and global graph features to predict atom/bond edits and a termination, respectively. Instead of adding a atom or benzene to the graph, we attach the subgraphs, called leaving groups, to the intermediates to complete reactants generation. This can significantly reduce the length of graph editing and further enhance predictive performance.

(c)  Rather than only considering the changes in atomic hydrogens and bond type between product and reactants, we refine the edit labels by introducing chirality and cis-trans isomerism in predefined atom and bond edits in an attempt to predict the stereochemistry of certain reactions.

There still remain certain challenges for the widespread application of Graph2Edits. First, the model cannot handle attaching the same leaving group to more than one atom in a molecular graph as there is

no bond addition in the predefined edits. A typical example is the reaction of protecting a carbonyl or aldehyde group to a cyclic acetal (Supplementary Fig. 2). Additionally, extraction of graph edits from datasets is highly reliant on atom-mapping information between products and reactants, which means incorrect matches would generate misleading edit sequences that bias the trained model. It should be mentioned that due to the lack of reaction conditions, there may be some gaps between the reaction generation process predicted by our model and the actual chemical reaction mechanism in the generation order or other details. And because of this, our model can provide a variety of reactants for target compound based on the frequency of reaction transformation rules in the training set, as the retrosynthesis is a one-to-many mapping problem and there might be several different reaction pathways to synthesize the target compound. Thus, this challenge can prompt us to design AI retrosynthesis model closer to chemical knowledge in the near future. Furthermore, although a target compound may have multiple reaction centers and produce diverse substrates through different reaction types, its reactivity may be specific to unique chemical environments. Future work on introducing more chemically meaningful modules and collecting high-quality reaction datasets will allow to better boost the applicability and interpretability of the model for the single-step retrosynthesis prediction.

## Methods
### Details of graph edits
Our graph edits are derived from the training set and represent the process of graph transformations in the retro-reactions. Since each atom is mapped on product and reactants, we mark the edit atoms or bonds to specify the positions and changes in each reaction. There are four different types of edits in reactions: (1) Delete bond, (2) Change bond, (3) Change atom, (4) Attach leaving group (LG) on atom, and our priority order for graph edits is Delete bond > Change bond > Change atom > Attach LG. The examples of edits derived from reactions are shown in Supplementary Fig. 1.

As shown in Supplementary Fig. 1a, the first edit is ('Change Atom', (0, 0)) on the atom 1, and the two numbers in brackets represent the number of hydrogen and the chiral type to be changed, respectively. And then, the graph edits is ('Attaching LG', '*C(=O)c1ccccc1C(*)=O') represents the '*C(=O)c1ccccc1C(*)=O' is added on the atom 1. At the reaction shown in Supplementary Fig. 1b, the bond [2, 3] is deleted and then the '*N(C)C' and '*Br' group are added on the atom 2 and 3, respectively. At the bottom of Supplementary Fig. 1c, the edit is first to delete the bond [6, 7] and [10, 11], this sequence may not match the true reaction mechanism, but it does not affect the final result of the graph transformation. Next, the bond edits ('Change Bond', (2, 0)), ('Change Bond', (1, 0)), ('Change Bond', (1, 0)) are operated on bond [7, 8], [8, 10], [6, 11], and the two numbers in brackets of bond edit represent the bond type and the bond stereo configuration to be changed. Finally, the leaving group '*=O' and '*Br' are attached on the atom 11 and 6, respectively.

There are also some incorrect graph edits sequence which derived from a small number of reactions using our automatic preprocessing method. The examples are shown in Supplementary Fig. 2. A common feature of these reactions is that the same leaving group needs to be added to more than one atom. And since there is no bond addition in the predefined edits, our method cannot handle this. Fortunately, there is little reactions of new bond formation in the training set (about 0.1%)[16].

After generating the ground truth edits sequence based on the atom mapping in reactions, we build the edits vocabulary. All graph edits were derived from the training set of USPTO-50k dataset, including 6 bond edits, 152 atom edits (7 Change Atom and 145 Attach LG), and a termination symbol and the details can be seen in Supplementary Data 1. The same procedure was used to build the edits vocabulary on USPTO-full dataset and the difference is that the edits

Attach LG must appear at least 50 times in the training set of USPTO-full before it will be collected into the vocabulary. This edits vocabulary include 6 bond edits, 336 atom edits (8 Change Atom and 328 Attach LG), and a termination symbol.

### Input representation
Given a compound, we represent it as a molecular graph $\mathcal{G} = (\mathcal{V}, \mathcal{E})$, where vertices $\mathcal{V}$ and edges $\varepsilon$ are atoms and bonds. Each node $v_i \in \mathcal{V}$ has a corresponding feature vector $x_i$ and each edge $e_{ij} \in \mathcal{E}$ has a feature vector $x_{ij}$. The initial features used for atoms and bonds can be found in the Supplementary Table 6 and 7.

### Graph encoder
The MPNN is a framework for multi-layer spatial convolutional GNNs, which operates on an undirected graph $\mathcal{G}$ to build the atom representations of molecule. Each layer comprises two main components, namely, message passing (Eq. (1)) and update (Eq. (2)):

$$m_i^{l+1} = \sum_{v_j \in N(v_i)} M_l\left(h_i^{(l)}, h_j^{(l)}, e_{ij}\right) \tag{1}$$

$$h_i^{(l+1)} = U_l\left(h_i^{(l)}, m_i^{(l+1)}\right) \tag{2}$$

where $N(v_i)$ denotes a set of neighbors of a given atom $v_i$. In short, at iteration/layer $l$, node messages $m_i^{(l)}$ and hidden states $h_i^{(l)}$ associated with each node $v_i$ are updated using the message function $M_l$ and node update function $U_l$. This has the effect that at each iteration, a node would be updated with the features from all of its adjacent nodes. However, such a mechanism is likely to introduce noise into the graph representation (a node message can appear more than once in a path)[52,55,65].

Here, in order to avoid the redundancy in node messages passing with MPNN, we base our work on the D-MPNN, which propagates messages along directed edges instead of nodes. And the corresponding message passing update equations are as follows:

$$m_{ij}^{(l+1)} = \sum_{v_k \in N(v_i) \setminus v_j} M_l\left(v_i, v_k, h_{ki}^{(l)}\right) \tag{3}$$

$$h_{ij}^{(l+1)} = U_t\left(h_{ij}^{(l)}, m_{ij}^{(l+1)}\right) \tag{4}$$

Note that $h_{ij}^{(l)}$ and $m_{ij}^{(l)}$ are distinct from $h_{ji}^{(l)}$ and $m_{ji}^{(l)}$, where the former are feature vectors along the edge $e_{i \to j}$ while the latter are feature vectors along the edge $e_{j \to i}$. And to update edge $e_{i \to j}$, Eq. (4) passes messages from its neighboring edges $e_{k \to i}$ that do not contain the edge $e_{j \to i}$ (the opposite direction to $e_{i \to j}$), ensuring that information only flows in one direction and reducing redundancy. We implement the message passing functions $M_l$ and edge update functions $U_l$ as follows:

$$M_l\left(v_i, v_j, h_{ij}^{(l)}\right) = h_{ij}^{(l)} \tag{5}$$

$$U_l\left(h_{ij}^{(l)}, m_{ij}^{(l+1)}\right) = GRU\left(h_{ij}^{(0)} + m_{ij}^{(l+1)}\right) \tag{6}$$

Prior to the first step of message passing, we initialize edge hidden states according to

$$h_{ij}^{(0)} = W_i(x_i \| x_{ij}) \tag{7}$$

where $W_i$ is a learnable weight matrix, $\|$ refers to concatenation operation. After the final iteration $L$ of edge features updates, the atom $v_i$ is represented as the aggregation of all the incoming bonds

features via:

$$h_i = \sigma\left(W_o\left(x_i \parallel \sum_{v_j \in N(v_i)} h_{ji}^{(L)}\right) + c\right) \tag{8}$$

where $W_o$ is the weights and $c$ is the bias of the fully connected layer, $\sigma$ stands for the ReLU activation function.

## Graph edits sequence generation

For given a product $\mathcal{G}_p$, Graph2Edits first autoregressively generates a sequence of edits $(e_1, \ldots, e_T)$, and then applies them to infer intermediates $\mathcal{G}_m$ sequentially until the final reactants $\mathcal{G}_r$ are obtained. At each generation step $t$, we take the intermediate graph $\mathcal{G}_m^{(t)}$ (in the first-generation step, $\mathcal{G}_m^{(1)} = \mathcal{G}_p$) as input and obtain the atom hidden states $h_i^{(t)}$ by D-MPNN encoder. To enhance the connection between the generation steps, we incorporate the previous step representations into current atom features via:

$$h_i^{(t)} = \sigma\left(W_v h_i^{(t-1)} + W_c h_i^{(t)}\right) \tag{9}$$

Since the number of atoms changes after attaching the leaving group, we zero-pad features of $h_i^{(t-1)}$ for any atom that was added to the graph at step $t$. After the atom features are updated, the bond features are represented by concatenating two atom features as

$$h_{ij}^{(t)} = \left(h_i^{(t)} \parallel h_j^{(t)}\right) \tag{10}$$

And we sum the atom hidden states to obtain a feature vector for the molecule

$$h_{\mathcal{G}}^{(t)} = \sum_{v_i \in \mathcal{G}_m^{(t)}} h_i^{(t)} \tag{11}$$

Finally, the logits $s_{(ij,b)}^{(t)}$, $s_{(i,a)}^{(t)}$ and $s_{\mathcal{G}}^{(t)}$ for bond edits $b \in E_{bond}$, atom edits $a \in E_{atom}$ and termination symbol are calculated at each step $t$ through the fully connected layers

$$s_{(ij,b)}^{(t)} = u_b{}^T(\sigma(W_b h_{ij}^{(t)} + c_b)) \tag{12}$$

$$s_{(i,a)}^{(t)} = u_a{}^T(\sigma(W_a h_i^{(t)} + c_a)) \tag{13}$$

$$s_{\mathcal{G}}^{(t)} = u_{\mathcal{G}}{}^T(\sigma(W_{\mathcal{G}} h_{\mathcal{G}}^{(t)} + c_{\mathcal{G}})) \tag{14}$$

where $u_b$ and $W_b$ are the weights and $c_b$ is the bias of bond edits predictor, $u_a$ and $W_a$ are the weights and $c_a$ is the bias of atom edits predictor, $u_{\mathcal{G}}$ and $W_{\mathcal{G}}$ are the weights and $c_{\mathcal{G}}$ is the bias of termination predictor.

## Training

We utilize teacher forcing[66] to train the model, that is, to predict each step edits during graph generation, we use previous steps from the ground-truth as input to the model. At each edit step $t$, each bond $e_{ij}$ in $\mathcal{G}_m^{(t)}$ has a label $y_{(ij,b)}^{(t)} \in \{0,1\}$, each atom $v_i$ is associated with a label $y_{(i,a)}^{(t)} \in \{0,1\}$ and graph label $y_{\mathcal{G}}^{(t)} \in \{0,1\}$. The optimization goal for prediction is to minimize the cross-entropy loss over possible edits,

aggregated over edit steps

$$\begin{aligned}\mathcal{L} = -\sum_{t \in T} \sum_{(\mathcal{G}_m, E)} &\left( \sum_{b \in E_{bond}} y_{(ij,b)}^{(t)} \log\left(s_{(ij,b)}^{(t)}\right)\right.\\ &\left.+ \sum_{a \in E_{atom}} y_{(i,a)}^{(t)} \log\left(s_{(i,a)}^{(t)}\right) + y_{\mathcal{G}}^{(t)} \log\left(s_{\mathcal{G}}^{(t)}\right)\right)\end{aligned} \tag{15}$$

Our model is implemented in PyTorch[67]. We also use the open-source software RDKit[50] to canonicalize product molecules, extract edits from reactions, attach leaving groups to intermediates and generate reactant SMILES.

## Evaluation and applying edits

We use beam search[68] with a Softmax scoring function to generate multiple ranked candidates for each product. During the generation process, we set the maximum number of steps to 9 and the beam width $k$ to 10. And at the step $(t)^{th}$, for a beam width $k$, we first calculate the probabilities of all possible edits and select $k$ edits with highest scores, then apply them to the input graph to obtain $k$ intermediates. Once this is done, the top $k$ intermediates graphs among all the generated $k^2$ graphs in the $(t+1)^{th}$ generation step are selected as the input graphs for the next step. During the beam search, a generation branch will stop if step $t$ reaches the maximum step or the graph representation $s_{\mathcal{G}}^{(t)}$ indicates a termination. Finally, the top $k$ edits sequence and graphs, ranked by their likelihoods, will be collected as the final predictions. Notably, Given the input product and edits sequence in the test set, we can deduce the reactants by RDKit with 99.6% accuracy.

## Model implementation details

Model trainings use the Adam optimizer for gradient decent optimization and the initial learning rate is set to 0.001 (0.0001 for USPTO-full dataset) and controlled by learning rate decay. The learning rate decay would monitor the validation accuracy and reduce the learning rate by multiplying a factor of 0.8 when the accuracy reached a plateau (a threshold value for improvement set to 0.01) within a patience of 5 epochs. Model gradients are clipped at maximum norm of 10. The hidden dimension of the D-MPNN is set to 256, and each node is updated for 10 iterations by message passing and node embeddings is dropout with a probability of 0.15. We use the fully-connected layers with hidden dimension 512 and dropout rate 0.2 for predicting the initial edit scores. We train our models for 150 epochs with a batch size 32. All modeling experiments on USPTO-50k were carried out in about 20-24 hours (15 days for training on USPTO-full) on a single NVIDIA RTX 2060 GPU.

## Reporting summary

Further information on research design is available in the Nature Portfolio Reporting Summary linked to this article.

# Data availability

The data and predictions that support the results of this study are available at the Graph2Edits GitHub repo: https://github.com/Jamson-Zhong/Graph2Edits. Source data are provided with this paper.

# Code availability

The source code of this work and associated trained models are available at the Graph2Edits GitHub repo: https://github.com/Jamson-Zhong/Graph2Edits[69,70].

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

## Acknowledgements

This work was supported by the National Natural Science Foundation of China (Grant No. 62176272), Research and Development Program of Guangzhou Science and Technology Bureau (No. 2023B01J1016), Key-Area Research and Development Program of Guangdong Province (No. 2020B1111100001), and China Medical University Hospital (DMR-112-085).

## Author contributions

W.Z. and C.Y.-C.C. designed research. W.Z. and Z.Y. worked together to complete the experiment and analyze the data. W.Z., Z.Y. and C.Y.-C.C. wrote the manuscript together. All authors reviewed and approved the final manuscript.

## Competing interests

The authors declare no competing interests.
