## [Peer Review File · Nature Communications]

Retrosynthesis prediction using an end-to-end graph generative architecture for molecular graph editingREVIEWER COMMENTS

Reviewer #1 (Remarks to the Author):

The manuscript by Zhong et al. subjected to my revision describes a retrosynthetic model which is based on the flow of electrons in the reaction mechanism. In general, I like the organization of the paper, the results are important, the methodology is reasonable and described well, the figures are helpful and of good quality, and finally, I did not find any typo or grammar error, apart from maybe line 47, where I would change "synthetic routes of a target molecule" to "synthetic routes TO a target molecule".

Yet I think that the manuscript requires some more in-depth analysis of the possible problems of the method and a comparison to the literature.

First of all, there was already work by Bradshaw et al. ("A generative model for electron paths") in which a very similar concept was described, with even higher TopK accuracy. I do not claim, that the authors must therefore retract their paper, but for sure some comparison has to be done. Also, there are works on the prediction of reaction mechanism (e.g. Deep learning for chemical reaction prediction), which should be mentioned in my opinion.

Moreover, as far as I understood, the model is trained on USPTO-50k. This is a rather small dataset with only a limited number of reaction classes. In particular, many "hard" to analyze reactions may not be included in the analysis. Therefore, I guess that the model may have problems with pericyclic reactions, eliminations, or hydrogenations. If the authors can show, that the models can deal with previously unseen reactions, that would be beneficial for the model.

Moreover, training the model with the deep assumption, that the actions of the model somehow correspond to the reaction mechanism is rather dangerous, as the mechanism depends on the conditions. In particular, the chemistry-aware model should change the reaction order depending on the temperature, or at least on catalysts. USPTO-50k does not have information about conditions, the solvent, or catalysts, as far as I know, so I think that the model, although performing well, is still far from understanding the underlying chemistry. Again I think, that some discussion of the selection of the pathways in the absence of reaction conditions is necessary if the authors want to keep a chemist's viewpoint on the reaction.

Also, in my opinion, some numbers do not much. In particular, at least for RetroXpert the TopK values do not match those in the cited paper. It would be nice to have some clarification.

Furthermore, some of the papers (for sure "Permutation Invariant Graph-to-Sequence Model for Template-Free Retrosynthesis and Reaction Prediction", but maybe more) which were stored as ArXiv preprints are now already published. This may be included in the references.

Regarding figures, in general, I really like them, however, I think that in Fig. 1b I would specify where to add the LGs (attach LG *Br to C:3, etc). In the 1c lower panel, the description has some failures. Point 4 - there should be N:10 instead of C:10, there are two "attach *O", but "attach *Br" is missing, after attaching *O one has also to change the bond C:6-attached O to double. In Fig. 2 there is a positive charge on oxygen, which is not wrong, but rather non-standard. Also, I guess that if that was the case, the movement of electrons to the adjacent carbon atom would be hindered. Usually, the positive charge is on the Phosphorus atom, which allows the oxygen to pull the electrons also from the adjacent carbon forming there partial positive charge and allowing attack.

To sum up, I really like the work and think it was done with great care. Still, I feel that some topics should be discussed in more detail and some small things clarified.

Reviewer #2 (Remarks to the Author):

The authors proposed a retrosynthesis model named Graph2Edits to predict possible reactants from a given target molecule by performing sequential subgraph-level edits. While the model shows interesting 55.1% top-1 accuracy on a public reaction dataset, I have some major concerns.

Comments:

1. I don't see the connection between chemical mechanism (arrow-pushing) and the proposed method. This method does not look different from the previous product-synthon-reactant prediction pipelines. The figure 2 should show the relationship between reaction mechanism and this method if the authors want to emphasize the relationship. Take the Mitsunobu reaction in Figure 2 as an example, the leaving group is O=PPh₃, which the phosphine comes from the reagent PPh₃. I believe the prediction does not include O=PPh₃ but simply break the C-C bond and add OH group on one side.
2. This work is very similar to MEGAN (figure 3 in this manuscript and figure 1 in the MEGAN paper are almost identical), where the current authors change the action space from atom-level to leaving group-level, and in this sense, the current method may be seen a semi-template-based version of MEGAN. I suggest the authors spend more words on comparing the proposed method with MEGAN, rather than describing this method as a new concept inspired by reaction mechanism. (The semi-template-based method is defined by predefining chemical fragments to perform retrosynthesis. Therefore, RetroPrime and MEGAN should be categorized into template-free method, as none of them pre-defined chemical fragments.)
3. In line 171, how is the edits vocabulary E built?
4. To show the prediction accuracy of stereochemistry, why don't the authors show only the top-k accuracy of those 75 reactions with stereochemical change? It's not clear to see the results if those predictions are mixed up.
5. What's the point of visualizing the graph embedding by t-SNE plot? Isn't it normal to see data clustering after training more epochs? I don't really see much scientific insight here.
6. In the conclusion, the authors claimed they "verified the superiority of the proposed method in complex stereochemistry and rare reactions" and "more efficiently search the latent space of the plausible reactions, and improve the variety of prediction results and interpretability". These claims should be supported with more experimental evidences. I cannot agree with any of these conclusions in the current manuscript. Another example, the variety of prediction, I don't see any comment on this through the manuscript unless I missed it.
7. I am curious about the top-50 exact match accuracy of USPTO-50K dataset and those failed top-50 predictions. As the method is shown to cover 99.6% of the test set, it would be useful if authors can discuss the reason for those failed predictions and the potential direction to improve the model performance.
8. USPTO-50k is a small dataset for learning chemical reactions, and it is essential that the authors show the results on a larger dataset, such as USPTO-MIT dataset. In addition, it would be useful to show a few multi-step retrosynthesis examples compared to literatures to show the practical use of proposed retrosynthetic model.

Reviewer #3 (Remarks to the Author):

The authors present an approach for AI-based single-step retrosynthesis relying on a semi-template method. The approach is interesting even if it still relies on the knowledge of the templates for training. The most interesting part is the autoregressive prediction of the edits, which mimics the “reaction mechanism”, making the single-step retrosynthesis models more interpretable.

Specific Comments:

1. The authors evaluate the model on USPTO 50k, which is quite a limited dataset. An evaluation on USPTO full would be more insightful.
2. The metric used to evaluate the models is topk. While widely used in the AI-retrosynthesis community, is not the most appropriate metric for the problem. As the authors point out as well this is a one-to-many problem.
3. Pay attention to English in the manuscript (e.g. Line 119-125), and be more precise in the sentences (e.g. line 45 is not Coley that first proposed retrosynthesis – this is something that has been around for much longer, it was just not handled yet with AI.)
4. I believe more details are needed on how the edits are extracted from the atom mapping (perhaps a figure?). Moreover, I believe that not for all reactions these can be identified with the reaction mechanism. In many (some?) cases the edits are not the true reaction mechanism, but just an artifact.
5. The authors claim that their model improves on rare reactions. However, with such a small dataset, split randomly, is hard to make such a claim. The authors would first need to check for the same rare reactions they extract, how the state-of-the-art performs and possibly also use a better split (ex. Tanimoto-based splitting).
6. The authors claim that template-free approaches output many invalid predictions. However, I guess also their model can generate “invalid edits” that do not bring to a valid chemical structure. Can this be quantified? And possibly compared to the topk percentage of invalid SMILES generated by state-of-the-art template-free approaches?

REVIEWER COMMENTS

Reviewer #1 (Remarks to the Author):

The manuscript by Zhong et al. subjected to my revision describes a retrosynthetic model which is based on the flow of electrons in the reaction mechanism. In general, I like the organization of the paper, the results are important, the methodology is reasonable and described well, the figures are helpful and of good quality, and finally, I did not find any typo or grammar error, apart from maybe line 47, where I would change "synthetic routes of a target molecule" to "synthetic routes TO a target molecule".

Yet I think that the manuscript requires some more in-depth analysis of the possible problems of the method and a comparison to the literature.

First of all, there was already work by Bradshaw et al. ("A generative model for electron paths") in which a very similar concept was described, with even higher TopK accuracy. I do not claim, that the authors must therefore retract their paper, but for sure some comparison has to be done. Also, there are works on the prediction of reaction mechanism (e.g. Deep learning for chemical reaction prediction), which should be mentioned in my opinion.

Moreover, as far as I understood, the model is trained on USPTO-50k. This is a rather small dataset with only a limited number of reaction classes. In particular, many "hard" to analyze reactions may not be included in the analysis. Therefore, I guess that the model may have problems with pericyclic reactions, eliminations, or hydrogenations. If the authors can show, that the models can deal with previously unseen reactions, that would be beneficial for the model.

Moreover, training the model with the deep assumption, that the actions of the model

somehow correspond to the reaction mechanism is rather dangerous, as the mechanism depends on the conditions. In particular, the chemistry-aware model should change the reaction order depending on the temperature, or at least on catalysts. USPTO-50k does not have information about conditions, the solvent, or catalysts, as far as I know, so I think that the model, although performing well, is still far from understanding the underlying chemistry. Again I think, that some discussion of the selection of the pathways in the absence of reaction conditions is necessary if the authors want to keep a chemist's viewpoint on the reaction.

Also, in my opinion, some numbers do not match. In particular, at least for RetroXpert the TopK values do not match those in the cited paper. It would be nice to have some clarification.

Furthermore, some of the papers (for sure "Permutation Invariant Graph-to-Sequence Model for Template-Free Retrosynthesis and Reaction Prediction", but maybe more) which were stored as ArXiv preprints are now already published. This may be included in the references.

Regarding figures, in general, I really like them, however, I think that in Fig. 1b I would specify where to add the LGs (attach LG *Br to C:3, etc). In the 1c lower panel, the description has some failures. Point 4 - there should be N:10 instead of C:10, there are two "attach *O", but "attach *Br" is missing, after attaching *O one has also to change the bond C:6-attached O to double. In Fig. 2 there is a positive charge on oxygen, which is not wrong, but rather non-standard. Also, I guess that if that was the case, the movement of electrons to the adjacent carbon atom would be hindered. Usually, the positive charge is on the Phosphorus atom, which allows the oxygen to pull the electrons also from the adjacent carbon forming there partial positive charge and allowing attack.

To sum up, I really like the work and think it was done with great care. Still, I feel that

some topics should be discussed in more detail and some small things clarified.

Reviewer #2 (Remarks to the Author):

The authors proposed a retrosynthesis model named Graph2Edits to predict possible reactants from a given target molecule by performing sequential subgraph-level edits. While the model shows interesting 55.1% top-1 accuracy on a public reaction dataset, I have some major concerns.

Comments:

1. I don't see the connection between chemical mechanism (arrow-pushing) and the proposed method. This method does not look different from the previous product-synthon-reactant prediction pipelines. The figure 2 should show the relationship between reaction mechanism and this method if the authors want to emphasize the relationship. Take the Mitsunobu reaction in Figure 2 as an example, the leaving group is O=PPh₃, which the phosphine comes from the reagent PPh₃. I believe the prediction does not include O=PPh₃ but simply break the C-C bond and add OH group on one side.
2. This work is very similar to MEGAN (figure 3 in this manuscript and figure 1 in the MEGAN paper are almost identical), where the current authors change the action space from atom-level to leaving group-level, and in this sense, the current method may be seen a semi-template-based version of MEGAN. I suggest the authors spend more words on comparing the proposed method with MEGAN, rather than describing this method as a new concept inspired by reaction mechanism. (The semi-template-based method is defined by predefining chemical fragments to perform retrosynthesis. Therefore, RetroPrime and MEGAN should be categorized into template-free method, as none of them pre-defined chemical fragments.)

3. In line 171, how is the edits vocabulary E built?
4. To show the prediction accuracy of stereochemistry, why don't the authors show only the top-k accuracy of those 75 reactions with stereochemical change? It's not clear to see the results if those predictions are mixed up.
5. What's the point of visualizing the graph embedding by t-SNE plot? Isn't it normal to see data clustering after training more epochs? I don't really see much scientific insight here.
6. In the conclusion, the authors claimed they "verified the superiority of the proposed method in complex stereochemistry and rare reactions" and "more efficiently search the latent space of the plausible reactions, and improve the variety of prediction results and interpretability". These claims should be supported with more experimental evidences. I cannot agree with any of these conclusions in the current manuscript. Another example, the variety of prediction, I don't see any comment on this through the manuscript unless I missed it.
7. I am curious about the top-50 exact match accuracy of USPTO-50K dataset and those failed top-50 predictions. As the method is shown to cover 99.6% of the test set, it would be useful if authors can discuss the reason for those failed predictions and the potential direction to improve the model performance.
8. USPTO-50k is a small dataset for learning chemical reactions, and it is essential that the authors show the results on a larger dataset, such as USPTO-MIT dataset. In addition, it would be useful to show a few multi-step retrosynthesis examples compared to literatures to show the practical use of proposed retrosynthetic model.

Reviewer #3 (Remarks to the Author):

The authors present an approach for AI-based single-step retrosynthesis relying on a semi-template method. The approach is interesting even if it still relies on the knowledge of the templates for training. The most interesting part is the autoregressive

prediction of the edits, which mimics the “reaction mechanism”, making the single-step retrosynthesis models more interpretable.

Specific Comments:

1. The authors evaluate the model on USPTO 50k, which is quite a limited dataset. An evaluation on USPTO full would be more insightful.

2. The metric used to evaluate the models is topk. While widely used in the AI-retrosynthesis community, is not the most appropriate metric for the problem. As the authors point out as well this is a one-to-many problem.

3. Pay attention to English in the manuscript (e.g. Line 119-125), and be more precise in the sentences (e.g. line 45 is not Coley that first proposed retrosynthesis – this is something that has been around for much longer, it was just not handled yet with AI.)

4. I believe more details are needed on how the edits are extracted from the atom mapping (perhaps a figure?). Moreover, I believe that not for all reactions these can be identified with the reaction mechanism. In many (some?) cases the edits are not the true reaction mechanism, but just an artifact.

5. The authors claim that their model improves on rare reactions. However, with such a small dataset, split randomly, is hard to make such a claim. The authors would first need to check for the same rare reactions they extract, how the state-of-the-art performs and possibly also use a better split (ex. Tanimoto-based splitting).

6. The authors claim that template-free approaches output many invalid predictions. However, I guess also their model can generate “invalid edits” that do not bring to a valid chemical structure. Can this be quantified? And possibly compared to the topk percentage of invalid SMILES generated by state-of-the-art template-free approaches?

Reviewer 1

The manuscript by Zhong et al. subjected to my revision describes a retrosynthetic model which is based on the flow of electrons in the reaction mechanism. In general, I like the organization of the paper, the results are important, the methodology is reasonable and described well, the figures are helpful and of good quality, and finally, I did not find any typo or grammar error, apart from maybe line 47, where I would change "synthetic routes of a target molecule" to "synthetic routes TO a target molecule". Yet I think that the manuscript requires some more in-depth analysis of the possible problems of the method and a comparison to the literature.

Reply: We sincerely thank the reviewer for the thorough review, constructive comments, and very useful suggestions. It is very helpful to improve the quality of our manuscript. According to your suggestions, we have revised the "synthetic routes of a target molecule" to "synthetic routes to a target molecule", and have made corrections in this revised manuscript, the detailed corrections are as follows.

Reviewer Point P1.1 — First of all, there was already work by Bradshaw et al. ("A generative model for electron paths") in which a very similar concept was described, with even higher TopK accuracy. I do not claim, that the authors must therefore retract their paper, but for sure some comparison has to be done. Also, there are works on the prediction of reaction mechanism (e.g. Deep learning for chemical reaction prediction), which should be mentioned in my opinion.

Reply: We thank the reviewer for this valuable comment. We do agree the reviewer's suggestion, indicating the importance of some works on the reaction mechanism prediction and the research basis of our work. Actually, predicting chemical reactions generally involves two mapping directions, predicting reaction outcomes for given reactants (forward synthesis prediction) or proposing possible reactants given the desirable product (retrosynthesis prediction). The related works mentioned by the reviewer belong to the forward synthesis prediction. A forward prediction may be

generally more straightforward, because the desired task is one-to-one mapping, that is, for given reactants and conditions, the reaction product is usually uniquely defined. In contrast, retrosynthesis is a one-to-many mapping, which is more challenging in the sense that there might be several different reaction pathways to synthesize the target product. Therefore, the top- k accuracy of forward prediction is generally much higher than that of retrosynthesis prediction and these two tasks are not comparable. To address this issue, we have revised and added a paragraph in the **Introduction Section** on page 6-7 lines 117-134 to describe the related work applying the similar concept.

“In organic synthesis, it is crucial to understand the reaction mechanism by applying the arrow-pushing approach which simplifies the stepwise electrons shift using sequences of arrows in molecular graphs¹. As shown in Fig. 2, a simplified mechanism example in the Mitsunobu reaction: the reagent PPh₃ (triphenylphosphine) combines with DEAD (diethyl azodicarboxylate) to generate a phosphonium intermediate that binds to the alcohol oxygen, activating it as a leaving group, then the nucleophile oxygen anion and the phosphonium ion to perform nucleophilic substitution to yield the final product. Based on the approximate reaction mechanism, there have been some machine learning models proposed for forward reaction prediction²⁻⁶. Bradshaw et al.² proposed a novel generative model for reaction mechanism prediction, which formulated the reaction electron paths as a sequence of graph transformations including bond removal and addition. Fooshee et al.³ also introduce a deep learning approach to predict and rank reaction outcomes through identifying electron sources and sinks. Similarly, GTPN⁴ integrated GNN and reinforcement learning (RL) to predict an optimal sequence of operation on atom pairs that transforms the reactants into products. However, most of these methods cannot be directly used for retrosynthetic prediction since no other leaving groups or atoms need to be added in the forward reaction prediction.”

Reviewer Point P1.2 — Moreover, as far as I understood, the model is trained on USPTO-50k. This is a rather small dataset with only a limited number of reaction classes. In particular, many "hard" to analyze reactions may not be included in the

analysis. Therefore, I guess that the model may have problems with pericyclic reactions, eliminations, or hydrogenations. If the authors can show, that the models can deal with previously unseen reactions, that would be beneficial for the model.

Reply: We thank the reviewer for this critical comment that indeed improved the quality of this manuscript. Admittedly, USPTO-50k is a small dataset with only 50016 reactions, but it is a clean dataset containing 10 reaction types commonly used by medicinal chemists and is widely used in previous retrosynthesis work⁷. To address this issue, we evaluated our model on the larger USPTO-full dataset to verify the scalability, and the performance result were shown in Table 3. Although the USPTO-full is much noisier than the clean USPTO-50k, our method, Graph2Edits, still achieved the top-1 accuracy of 44.0%, which is significantly higher than the baseline MEGAN, and is close to the state-of-the-art models. These results further demonstrate the scalability of the model. We have added Table 2 and two new paragraphs to the Section 2.1 and 2.2 on page 10-11 lines 213-216 and page 13 lines 264-271 in the revised manuscript.

“Additionally, we also use the original USPTO-full dataset from the entire USPTO (1976-Sep2016) to verify the scalability of our model. We use exactly the same splits as Dai et al.⁸, which contain approximately 800k/100k/100k training/validation/test reactions, and repeat the procedures given in the above USPTO-50k dataset processing.”

“We also compare the performance of Graph2Edits on the larger USPTO-full dataset with other baselines for retrosynthesis prediction. The results are presented in Table 3. Although the USPTO-full is much noisier than the clean USPTO-50k, our method still has competitive performance with a top-1 accuracy of 44.0%, on par with the semi-template-based method RetroPrime and outperforming MEGAN by a large margin. In addition, on larger k ($k > 1$), especially top-10 accuracy, Graph2Edits significantly outperforms all other methods except Aug.Transformer, showing similar superiority to the performance on the USPTO-50k dataset.”

Table 3 Top- k exact match accuracy of the proposed Graph2Edits and baselines on USPTO-full dataset.

Category	Model	Top- k accuracy (%)			
		$k=1$	3	5	10
Template-Based	Retrosim	32.8	-	-	56.1
	Neuralsym	35.8	-	-	60.8
	GLN	39.3	-	-	63.7
Template-Free	Aug. Transformer	46.2	-	-	73.3
	GTA	46.6	-	-	70.4
	Graph2SMILES	45.7	-	-	63.4
Semi-Template-Based	RetroPrime	44.1	59.1	62.8	68.5
	MEGAN	33.6	-	-	63.9
	Graph2Edits (D-MPNN)	44.0	60.9	66.8	72.5

As for whether our model can deal with some previously unseen reactions, we have presented some examples predicted by Graph2Edits in Fig. 7, including 1, 3-dipolar cycloaddition reaction, easter reduction reaction and amide dehydration elimination reaction, and the analysis for these predictions can be seen in Section 2.6. Moreover, we also provided the further analysis of incorrect predictions by our model in Supplementary Fig. 2 (Section 2.3). All these illustrate that our method can predict diverse reactants to synthesize the product.

Reviewer Point P1.3 — Moreover, training the model with the deep assumption, that the actions of the model somehow correspond to the reaction mechanism is rather dangerous, as the mechanism depends on the conditions. In particular, the chemistry-aware model should change the reaction order depending on the temperature, or at least on catalysts. USPTO-50k does not have information about conditions, the solvent, or catalysts, as far as I know, so I think that the model, although performing well, is still far from understanding the underlying chemistry. Again I think, that some discussion

of the selection of the pathways in the absence of reaction conditions is necessary if the authors want to keep a chemist's viewpoint on the reaction.

Reply: We appreciate the reviewer for pointing the important concern. We do agree with the reviewer that due to the lack of reaction conditions, there may be some gaps between the reaction generation process predicted by our model and the actual chemical reaction mechanism in the generation order or other details. Because of this, our model can provide a variety of reactants for the product based on the frequency of reaction transformation rules in the training set, as the retrosynthesis is a one-to-many mapping problem and there might be several different reaction pathways to synthesize the target compound. Although the predicted process may still far from understanding the underlying chemistry, our method provides an attempt for the AI model to understand the rules of chemical reaction transformation and perform retrosynthetic inference. To clearly describe this point, we have added a few discussions to the **Section 3 Discussion** on page 23 lines 490-497 as follows:

“It should be mentioned that due to the lack of reaction conditions, there may be some gaps between the reaction generation process predicted by our model and the actual chemical reaction mechanism in the generation order or other details. And because of this, our model can provide a variety of reactants for target compound based on the frequency of reaction transformation rules in the training set, as the retrosynthesis is a one-to-many mapping problem and there might be several different reaction pathways to synthesize the target compound. Thus, this challenge can prompt us to design AI retrosynthesis model closer to chemical knowledge in the near future.”

Reviewer Point P1.4 — Also, in my opinion, some numbers do not much. In particular, at least for RetroXpert the TopK values do not match those in the cited paper. It would be nice to have some clarification.

Reply: Thanks for your question. The authors in RetroXpert reported their performance being affected by the dataset leakage on their GitHub website (<https://github.com/uta-smile/RetroXpert>), Thus, we use the most recent results from their GitHub website on

the canonicalized dataset. To avoid misunderstanding, we added the following annotation to RetroXpert in Table 1:

^a"The results are taken from <https://github.com/uta-smile/RetroXpert>."

Reviewer Point P1.5 — Furthermore, some of the papers (for sure "Permutation Invariant Graph-to-Sequence Model for Template-Free Retrosynthesis and Reaction Prediction", but maybe more) which were stored as ArXiv preprints are now already published. This may be included in the references.

Reply: We appreciate the reviewer for pointing out the published reference. We checked all the references stored as ArXiv preprints carefully and the relevant published article has been revised as follow:

[30]Tu, Z. & Coley, C. W. Permutation invariant graph-to-sequence model for template-free retrosynthesis and reaction prediction. *Journal of chemical information and modeling* **12**, 3503-3513 (2022).

Reviewer Point P1.6 — Regarding figures, in general, I really like them, however, I think that in Fig. 1b I would specify where to add the LGs (attach LG *Br to C:3, etc). In the 1c lower panel, the description has some failures. Point 4 - there should be N:10 instead of C:10, there are two "attach *O", but "attach *Br" is missing, after attaching *O one has also to change the bond C:6-attached O to double. In Fig. 2 there is a positive charge on oxygen, which is not wrong, but rather non-standard. Also, I guess that if that was the case, the movement of electrons to the adjacent carbon atom would be hindered. Usually, the positive charge is on the Phosphorus atom, which allows the oxygen to pull the electrons also from the adjacent carbon forming there partial positive charge and allowing attack.

Reply: Thanks for your careful checks. We are really sorry for our careless mistakes. In revised Fig.1, we have added the atom-mapping information to the molecular structures and provided the additional edit sites (include bond and atom) to specify the location of graph edits, for example: in Fig. 1b, the bond [2, 3] is deleted and then the

'*N(C)C' and '*Br' group are added on the atom 2 and 3, respectively. In Fig.2, we agree with you that the positive charge on the Phosphorus atom is more LG, rational. The revised Fig.1 and Fig.2 are attached below for review.

a. Atom center reaction

b. Bond center reaction

c. Multiple centers reaction

Fig. 1 The examples of derived edits sequence in retro-reactions. **a** Atom center reaction, **b** Bond center reaction and **c** Multiple centers reaction. The two numbers in brackets of atom edit represent the number of hydrogen and the chiral type to be changed, and the two numbers in brackets of bond edit indicate the bond type and the bond stereo configuration to be changed. The changes in the reaction are highlighted in red.

Fig. 2 The motivation of the proposed method. An example of the arrow-pushing formalism in the mechanism reasoning process of Mitsunobu reaction. The steric configuration of the carbon atom in the reaction center is reversed. In the retrosynthesis prediction, DEAD (diethyl azodicarboxylate) and PPh₃ (triphenylphosphine) are two reaction reagents that were removed, and we regard the process of the simplified mechanism derivation as sequence of graph edits on the molecular structure.

Reviewer Point P1.7 — To sum up, I really like the work and think it was done with great care. Still, I feel that some topics should be discussed in more detail and some small things clarified.

Reply: We thank the reviewer for this valuable suggestion. To better reflect the quality of this work, we have rewritten the discussion section in more depth, and discussed the important aspects of the work, the challenges encountered in which the proposed method may fail and the direction of future work. More details have been added in Section Discussion on page 21-23 as follows:

“In this study, we developed a new end-to-end semi-template-based retrosynthesis prediction model, Graph2Edits, which predicts a possible sequence of edits from the product graph and sequentially generates the intermediates and reactants. In contrast to previous template-based methods that limit predictions to template sets and template-free models that fail to capture the rich structural information in molecular graph, Graph2Edits is a graph-based model that treats one-step retrosynthesis as applying a sequence of graph edits to product graph and generates reactant molecules just as

chemists think about how a reaction happened. Comprehensive evaluations on the benchmark dataset USPTO-50k demonstrate that our method achieves a promising 55.1% top-1 exact match accuracy and shows comparable or improved performance compared to other state-of-the-art models. In the large and noisy USPTO-full dataset, Graph2Edits also achieves the top-1 accuracy of 44.0%, which is significantly higher than the baseline MEGAN, and is close to the state-of-the-art models. These encouraging results display that our model has excellent generalization and robustness. Crucially, since the multi-step generation predicts arbitrary length edits, the model can more efficiently search the latent space of the plausible reactions and improve the diversity of prediction results. Extensive experiments verified the superiority of the proposed method in complex stereochemistry and rare reactions. In particular, detailed analyses of model predictions including molecular representations suggest that this strategy can enhance the rationality and interpretability of retrosynthetic models.

There still remain certain challenges for the widespread application of Graph2Edits. First, the model cannot handle attaching the same leaving group to more than one atom in a molecular graph as there is no bond addition in the predefined edits. A typical example is the reaction of protecting a carbonyl or aldehyde group to a cyclic acetal (Supplementary Fig. 1). Additionally, extraction of graph edits from datasets is highly reliant on atom-mapping information between products and reactants, which means incorrect matches would generate misleading edit sequences that bias the trained model. It should be mentioned that due to the lack of reaction conditions, there may be some gaps between the reaction generation process predicted by our model and the actual chemical reaction mechanism in the generation order or other details. And because of this, our model can provide a variety of reactants for target compound based on the frequency of reaction transformation rules in the training set, as the retrosynthesis is a one-to-many mapping problem and there might be several different reaction pathways to synthesize the target compound. Thus, this challenge can prompt us to design AI retrosynthesis model closer to chemical knowledge in the near future. Furthermore, predicted initial intermediates may be incorrect and thus not generate the true reactants. Although a target compound may have multiple reaction centers and produce diverse

substrates through different reaction types, its reactivity may be specific to unique chemical environments. Future work on introducing more chemically meaningful modules and collecting high-quality reaction datasets will allow to better boost the applicability and interpretability of the model for the single-step retrosynthesis prediction.

Reviewer 2

The authors proposed a retrosynthesis model named Graph2Edits to predict possible reactants from a given target molecule by performing sequential subgraph-level edits. While the model shows interesting 55.1% top-1 accuracy on a public reaction dataset, I have some major concerns.

Reply: We thank the reviewer for the valuable comments and suggestions. All the concerns raised from the reviewer have been addressed in detail as follows.

Reviewer Point P2.1 — I don't see the connection between chemical mechanism (arrow-pushing) and the proposed method. This method does not look different from the previous product-synthon-reactant prediction pipelines. The figure 2 should show the relationship between reaction mechanism and this method if the authors want to emphasize the relationship. Take the Mitsunobu reaction in Figure 2 as an example, the leaving group is O=PPh₃, which the phosphine comes from the reagent PPh₃. I believe the prediction does not include O=PPh₃ but simply break the C-C bond and add OH group on one side.

Reply: We appreciate the reviewer for the valuable comment. It should be mentioned that our method is different from the previous two-step product-synthon-reactant prediction methods, as these methods require training two separate models to complete the transformation, ignoring a strong link between center identification and synthon

completion in chemical reactions, and most of them only focus on at most one atom or bond center, making it challenging to deal with reactions involving several centers. Actually, in Fig.2, the reaction mechanism is described by applying the arrow-pushing approach which simplifies the stepwise electrons shift using sequences of arrows in molecular graphs, in contrast, retrosynthesis is the reverse of reaction and has no reaction conditions, thus, we simplified mechanism of retro-reaction transformations in some cases. To clearly express this relationship, we have added the description on the legend of Fig. 2 as follow:

“In the retrosynthesis prediction, DEAD (diethyl azodicarboxylate) and PPh₃ (triphenylphosphine) are two reaction reagents that were removed, and we regard the process of the simplified mechanism derivation as sequence of graph edits on the molecular structure.”

And inspired by sequences of arrows in mechanism, some forward reaction prediction models²⁻⁶ have been proposed by formulate a reaction as a sequence of graph edits such as bond removal or addition. Based on the successful application of these models and MEGAN, also similar motivated by the arrow-pushing mechanism, we proposed the Graph2Edits for predicting the graph edits of the product in an end-to-end architecture. Unlike the forward reaction prediction, the retrosynthesis is a one-to-many mapping problem and lack of reaction conditions, so there are some differences between the sequence of the graph edits and the actual chemical reaction mechanism in details. And we have revised the paragraph on page 6-7 lines 117-134 as follows:

“In organic synthesis, it is crucial to understand the reaction mechanism by applying the arrow-pushing approach which simplifies the stepwise electrons shift using sequences of arrows in molecular graphs¹. As shown in Fig. 2, a simplified mechanism example in the Mitsunobu reaction: the reagent PPh₃ (triphenylphosphine) combines with DEAD (diethyl azodicarboxylate) to generate a phosphonium intermediate that binds to the alcohol oxygen, activating it as a leaving group, then the nucleophile oxygen anion and the phosphonium ion to perform nucleophilic substitution to yield the final product. Based on the approximate reaction mechanism, there have been some machine learning models proposed for forward reaction prediction²⁻⁶. Bradshaw et al.²

proposed a novel generative model for reaction mechanism prediction, which formulated the reaction electron paths as a sequence of graph transformations including bond removal and addition. Fooshee et al.³ also introduce a deep learning approach to predict and rank reaction outcomes through identifying electron sources and sinks. Similarly, GTPN⁴ integrated GNN and reinforcement learning (RL) to predict an optimal sequence of operation on atom pairs that transforms the reactants into products. However, most of these methods cannot be directly used for retrosynthetic prediction since no other leaving groups or atoms need to be added in the forward reaction prediction.”

Reviewer Point P2.2 — This work is very similar to MEGAN (figure 3 in this manuscript and figure 1 in the MEGAN paper are almost identical), where the current authors change the action space from atom-level to leaving group-level, and in this sense, the current method may be seen a semi-template-based version of MEGAN. I suggest the authors spend more words on comparing the proposed method with MEGAN, rather than describing this method as a new concept inspired by reaction mechanism. (The semi-template-based method is defined by predefining chemical fragments to perform retrosynthesis. Therefore, RetroPrime and MEGAN should be categorized into template-free method, as none of them pre-defined chemical fragments.)

Reply: We deeply appreciate the reviewer’s comment, which helped us improve our manuscript. For Fig.3, although the models generated graph edits by the same auto-regression manner and have certain similarity in the process, in details, our architecture based on D-MPNN greatly simplified the encoder-decoder framework and molecular initial features processing, and the specific algorithm process is provided in Figure 3b. And also, the reaction example presented in Fig. 3a is Mitsunobu reaction, which is quite different from the one in MEGAN, and is more challenging for model to predict. Additionally, we agree the reviewer that based on MEGAN and also inspired by the arrow-pushing mechanism, our model changed the action space from atom level to

leaving group level since adding atoms results in a long prediction sequence which increases generation difficulty, and simplified some atomic editing to make it closer to the real chemical reaction. And also, we simplified the network architecture to effectively learn molecular representations and improved the efficiency on applying edits by RDKit for generating the reactants. To clarify this, we have added a comparison on page 7 lines 134-139 and page 8 lines 148-151 as follows:

“And it should be mentioned that the semi-template-based MEGAN⁶ was the first to model the reaction as an editing sequence for retrosynthesis prediction. Perhaps due to the complex encoder-decoder framework and the add operations at the atomic level, that work made the reactant generation challenging and performed not well in reactions that require attaching the large leaving group, and showed relatively low accuracy on benchmark dataset.”

“It is worth noting that unlike MEGAN model, we simplify the network architecture to effectively learn molecular representations, replace the add-atom actions with attaching substructures to reduce generation steps, and improve the efficiency for generating the reactants.”

And also, we apologized for not clearly describing the classification of the existing retrosynthesis models. Actually, the semi-template-based method is defined as not using a reaction template, nor directly converting the product to the reactant, but predicting the final reactant through the intermediates or synthons generated in multiple steps. And the RetroPrime and MEGAN also use predefined labels in the two-step workflow to predict synthons or intermediates, thus, we categorized into the semi-template-based method. To clarify this point, we have added a definition for the semi-template-based method on page 5 lines 98-100 as follows:

“The semi-template-based method is defined as not using a reaction template, nor directly converting the product to the reactant, but predicting the final reactant through the intermediates or synthons generated in multiple steps.”

Reviewer Point P2.3 — In line 171, how is the edits vocabulary E built?

Reply: Thank you for your question. We have responded to this comment along with our response to Reviewer Point P 3.4, please refer to our answer to P3.4.

Reviewer Point P2.4 — To show the prediction accuracy of stereochemistry, why don't the authors show only the top-k accuracy of those 75 reactions with stereochemical change? It's not clear to see the results if those predictions are mixed up.

Reply: We deeply appreciate the reviewer for the valuable suggestion. In order to clearly display the results of our model, we have recounted 74 (1.5%) reactions with stereochemical change in the test set of USPTO-50k and evaluated the top-*k* accuracy on these 74 reactions. We have shown the results in Fig. 5 and revised the relevant paragraph on page 15-16 lines 323-338 as follows:

“We count 74 (1.5%) reactions with stereochemical change in the test set of USPTO-50k and display the top-*k* accuracy on these reactions with or without stereochemistry information (including chirality and cis-trans isomerism) in Fig. 5. When removing the stereochemical information in the reactions, the accuracies of top-1 and top-10 reach 60.8% and 91.9% respectively, indicating that the model has deeply learned the transformation rules of these reactions and successfully predicted the true reactants. When with the stereochemical information in the reactions, although the performance drops obviously, the top-10 accuracy still achieves 66.2%. The reason may be that some reaction conversion rules do not have stereochemical changes, but are introduced by chiral catalysts or ligands. Therefore, our model tends to predict substrates with no stereo information (see incorrect predictions in Supplementary Fig. 2b). On the other hand, the high accuracy of top-10 is also due to the fact that our model performs well on reactions that inherently contain stereo changes (Fig. 6c). These results suggest that although the model may not be able to accurately predict the changes of stereochemistry

information in the reaction, it has an advantage in predicting stereochemical reactions and can learn some rules of stereochemical changes.”

Fig. 5 Top-k accuracy on 74 stereochemical reactions considering with/without stereochemistry. The blue and orange bars represent the performance of reactions with or without stereochemistry information.

Reviewer Point P2.5 — What’s the point of visualizing the graph embedding by t-SNE plot? Isn’t it normal to see data clustering after training more epochs? I don’t really see much scientific insight here.

Reply: We thank the reviewer for the valuable comment. At different stages of training, the molecular representations learned by the model are different, some of which are related to the predicted tasks, and some are not. For a specific prediction task, a good training process is that with the increase of epochs, the model continues to converge, and the learned features are more relevant to the task. In Fig.9, we visualized the molecular graph representations learned by our model at different training stages to show that our method can perceive the molecular characteristics on different edit steps

for retrosynthesis prediction. And we have also added a summary on this section (page 20 lines 422-423) as follows:

“These results suggest that our model can perceive the molecular characteristics on different edit steps for retrosynthesis prediction.”

Reviewer Point P2.6 — In the conclusion, the authors claimed they “verified the superiority of the proposed method in complex stereochemistry and rare reactions” and “more efficiently search the latent space of the plausible reactions, and improve the variety of prediction results and interpretability”. These claims should be supported with more experimental evidences. I cannot agree with any of these conclusions in the current manuscript. Another example, the variety of prediction, I don’t see any comment on this through the manuscript unless I missed it.

Reply: We thank the reviewer for pointing the important points. To address these issues, we have carried out more experiments to support our conclusions. For the first point, “the superiority of our method in complex stereochemistry”, we have evaluated the top-*k* accuracy on reactions with stereochemical change in the test set and clearly displayed the results in Fig.5, and these descriptions can be seen in Reviewer Point P 2.4. In addition, we have also analyzed the correct and incorrect predictions on stereochemical reactions in Fig. 6c and Supplementary Fig. 2b. All these results indicate that although the number of stereochemical reactions in the data set is small, our model has a certain advantage in predicting reactions that inherently contain stereo changes, and can learn some rules of stereochemical changes. For the second point, “the superiority of our method in rare reactions”, in the Section 2.4, we evaluated the performance of our model on reactions with the different edits sequence length and found that with the increase of edits sequence length, the number of reaction samples decreased dramatically, while the performance of our model did not decrease significantly. Thus, we concluded that our model remained relatively robust in rare reactions. Furthermore, we also investigated the performance on reactions with different reaction similarities

which based on Tanimoto coefficient to verify this conclusion, and we have added a new paragraph to the Section 2.4 page 15 lines 308-320 as follows:

“To further investigate the performance on reactions with different reaction similarities, we use Tanimoto coefficient⁹ to calculate the reaction similarities. For each reaction, we calculate the 2048-bit reaction fingerprints¹⁰, and the similarity is quantified by the mean Tanimoto similarity between the reaction and all other reactions in the test set. We cluster the reactions according to their reaction similarity using the K-means clustering algorithm in Euclidean distances¹¹. Table 4 presents the clustering results, and we can see that the clusters with high reaction similarity generally have higher top-*k* accuracy. It is worth noting that cluster 1, which has the lowest similarity with other reactions, can be regarded as a rare reaction cluster, and its top-1 accuracy can also reach nearly 50%. These results demonstrate that our performance is not obtained by overfitting to one particular category of reactions. Such a trait is also crucial since retrosynthesis may involve rare reactions that haven't been thoroughly explored in the literature.”

Table 4 Top-*k* exact match accuracy per each cluster with different reaction similarity.

Cluster	Cluster size	Mean similarity	Top- k accuracy (%)			
			k = 1	3	5	10
1	181	0.019	49.2	69.1	73.5	82.3
2	326	0.032	56.4	76.7	82.8	87.1
3	517	0.042	56.1	78.3	85.5	90.7
4	631	0.050	51.2	75.1	81.3	89.1
5	663	0.057	52.6	75.0	83.3	89.3
6	684	0.065	56.0	78.2	83.5	90.2
7	702	0.072	56.8	77.5	84.3	89.6
8	605	0.079	56.9	78.1	84.1	89.1
9	448	0.086	57.4	80.1	85.5	91.3
10	247	0.096	56.7	80.2	84.6	90.3

For the third point, “more efficiently search the latent space of the plausible reactions and the variety of prediction”, our model has mechanisms to efficiently search the possible chemical conversion process for reactions, and can continuously generate graph edits in an autoregressive manner, and output multiple different reaction centers and leaving groups in beam search, thus enabling the ability to predict reactants with different scaffolds and structures. To illustrate this point, we have shown some examples of diverse reactants predicted by Graph2Edits in Fig. 7, and quantitatively analyzed the diversity of predictive results based on the predicted reactants similarity in Fig. 8. We have added a new **Section 2.6** on page 17-19 lines 358-393 as follows:

“Diversity on predicted reactants

Evaluation the diversity of the predicted reactions is crucial, as it is related to whether the predictions of our method can cover a broad range of chemical reactions and enable novel discoveries in multi-step retrosynthetic route planning. Benefiting from our design strategy, Graph2Edits can continuously generate graph edits in an autoregressive manner, and output multiple different reaction centers and leaving groups in beam search, thus enabling the ability to predict reactants with different scaffolds and structures. To analyze the diversity of predicted results, we first present some examples of diverse reactants predicted by Graph2Edits in Fig. 7. The first example is 1, 3-dipolar cycloaddition reaction (Fig. 7a). Our model predicts four different reaction centers, including a nitrogen atom in triazole (top-1, 3, 4, 6 and 9), the whole triazole ring (top-2 and top-5) and two carbon-carbon bonds between aromatic rings (top-7 and top-8). And among these results, three reaction types (the amino protection with different protective groups, 1, 3-dipolar cycloaddition and the Suzuki cross-coupling reaction) are predicted to yield the product. In the second example (Fig. 7b), Graph2Edits

suggests a reduction of the ethyl ester or methyl ester (top-1 and top-2), which matches the ground-truth reaction. In addition, our method further offers the options of the hydroxyl protection and the aromatic coupling reaction. As depicted in Fig. 7c, for the reaction of the amide dehydration elimination to form the cyano group, our approach generates the ground-truth reactants in top-1 prediction, and can also provide the heterocycle formation, amino protection and double bond reduction with multiple distinct substrates.

To quantitatively analyze the diversity of predictive results, we investigate the molecular similarities among them. For each product, the similarity is quantified by the mean Tanimoto similarity between the predicted reactants and other top-10 predictions, based on the concatenated ECFP4 fingerprints, and the lower similarity indicates the higher diversity of predicted results. We also use the K-means clustering algorithm to cluster the products according to the similarity of predicted reactants, similar to that used by Chen et al.¹². As shown in Fig. 8, the first four clusters (dark red to orange) have lower prediction similarities (0.22, 0.36, 0.44 and 0.50), which can be regarded as high-diversity clusters, accounting for about 30% of the test set. The average similarity on middle three clusters (light orange and light blue) is 0.55, 0.60 and 0.65, respectively, and thus can be referred to as medium-diversity clusters, accounting for nearly 54% in test set. And the last three clusters (dark blue), considered as low-diversity clusters, have a small proportion and relatively higher prediction similarities (0.71, 0.80 and 0.98). All these results clearly show that Graph2Edits can predict diverse results.”

Fig. 7 Examples of predicted reactions by Graph2Edits. **a** 1, 3-dipolar cycloaddition, **b** Reduction reaction and **c** Elimination reaction.

Fig. 8 The cluster results on USPTO-50k test set based on predicted reactants similarities. The numbers above the bars represent the average similarity of the predicted reactants, and the lower the similarity, the higher the diversity.

And for the last point, “the interpretability”, Compared with the end-to-end template-free method to directly predict the reactants, our model predicted the reaction conversion steps in the way of auto-regression end-to-end, and outputted the final reactants through multi-step prediction. This process itself has a strong interpretability for the final predictions of the model as depicted in Section 2.5 and Fig. 6. In addition, we also visualized the molecular graph representations learned by our model at different training stages to show that our method can perceive the molecular characteristics on different edit steps (Section 2.7 and Fig. 9). These exciting results all demonstrate the great interpretability of our method for retrosynthesis prediction.

Reviewer Point P2.7 — I am curious about the top-50 exact match accuracy of USPTO-50K dataset and those failed top-50 predictions. As the method is shown to cover 99.6% of the test set, it would be useful if authors can discuss the reason for those failed predictions and the potential direction to improve the model performance.

Reply: We appreciate the reviewer for a helpful comment. We have calculated the top-50 exact match accuracy on USPTO-50K dataset and the results are shown on Table 1. Our model also achieved excellent performance on top-50 predictions and the accuracies have reached nearly 93% or more. In addition, we also analyzed the reactions that failed in top-50 predictions and the examples were shown in Supplementary Fig. 3. To discuss this issue, we have added a new Section 2.3 on page 13-14 as follows:

Table 1 Top- k exact match accuracy of the proposed Graph2Edits and baselines on USPTO-50K dataset.

Category	Model	Top- k accuracy (%)									
		Reaction class unknown					Reaction class known				
		$k = 1$	3	5	10	50	1	3	5	10	50
Template-Based	Retrosim	37.3	54.7	63.3	74.1	85.3	52.9	73.8	81.2	88.1	92.9
	Neuralsym	44.4	65.3	72.4	78.9	83.1	55.3	76.0	81.4	85.1	86.9
	GLN	52.5	69.0	75.6	83.7	92.4	64.2	79.1	85.2	90.0	93.2
	LocalRetro	53.4	77.5	85.9	92.4	97.7	63.9	86.8	92.4	96.3	97.9
Template-Free	SCROP	43.7	60.0	65.2	68.7	-	59.0	74.8	78.1	81.1	-
	Aug.Transformer	53.2	-	80.5	85.2	-	-	-	-	-	-
	GTA	51.1	67.6	74.8	81.6	-	-	-	-	-	-
	Graph2SMILES	52.9	66.5	70.0	72.9	-	-	-	-	-	-
	Dual-TF	53.6	70.7	74.6	77.0	-	65.7	81.9	84.7	85.9	-
Semi-Template-Based	G2G	48.9	67.6	72.5	75.5	-	61.0	81.3	86.0	88.7	-
	RetroXpert ^a	50.4	61.1	62.3	63.4	64.0	62.1	75.8	78.5	80.9	83.5
	RetroPrime	51.4	70.8	74.0	76.1	-	64.8	81.6	85.0	86.9	-
	MEGAN	48.1	70.7	78.4	86.1	93.2	60.7	82.0	87.5	91.6	95.3
	GraphRetro	53.7	68.3	72.2	75.5	-	63.9	81.5	85.2	88.1	-
	Graph2Edits (MPNN)	52.7	77.2	85.3	91.0	97.1	65.7	87.3	92.0	95.3	97.8
	Graph2Edits (D-MPNN)	55.1	77.3	83.4	89.4	92.7	67.1	87.5	91.5	93.8	94.6

^aThe results are taken from <https://github.com/uta-smile/RetroXpert>. Best results for each group are highlighted in bold.

Supplementary Fig. 2. Examples of failed reactions in top-50 predictions.

“Analysis of incorrect predictions

To more comprehensively understand performance, we perform an error analysis in top-50 predictions on the USPTO-50k test set. We visualize the top-10 predictions which are different from the ground truth reactants for two cases in Supplementary Fig. 2. We can observe that the common feature of these two products is to have multiple possible reaction centers, and thus can be yielded through a variety of different reaction types. In fact, all top-10 predicted reactants are feasible and can be synthesized by standard methods, although the reaction yields may vary. In Supplementary Fig. 2a, the correct reaction center is identified on top-3 and the top-7 prediction, but the subsequent predicted leaving group are incorrect. We speculate that this is due to the fact that the leaving group ‘*I’ attaching is less common in the training set than the leaving groups

'*Cl' and '*Br'. And in Supplementary Fig. 2b, the model identifies both the bond edit and leaving group correctly on top-2 prediction, but fails to predict the two chiral changes. This is also understandable, because the chirality in ground-truth reaction may be introduced by chiral ligand, and there is no change on molecular chirality in ordinary reactions. Based on these, there is still significant room to improve the model performance, for example, enhancing the ability of model to identify the reactivity of different reaction sites in the product to rank the predictions, and improving the chiral accuracy of the predicted reactants.”

Reviewer Point P2.8 — USPTO-50k is a small dataset for learning chemical reactions, and it is essential that the authors show the results on a larger dataset, such as USPTO-MIT dataset. In addition, it would be useful to show a few multi-step retrosynthesis examples compared to literatures to show the practical use of proposed retrosynthetic model.

Reply: We deeply appreciate the reviewer’s comment, which helped us improve our manuscript. We do agree with the reviewer that evaluating on a larger dataset is essential. We tried to conduct experiments on USPTO-MIT¹³ dataset, but found that although the dataset contains chiral information, there is no reaction in which the molecular chirality changes during data preprocessing. Thus, to address this issue, we evaluated our model on the larger and noisier USPTO-full⁸ dataset, and we have responded to this comment along with our response to Reviewer Point P 1.2, please refer to our answer to P 1.2. For multi-step retrosynthesis prediction, we have shown three examples, including the oral SARS-CoV-2 M^{pro} inhibitor Nirmatrelvir, the third-generation EGFR inhibitor Osimertinib and the Lenalidomide, to verify the practical use of our method, and we have added a new Section 2.8 on page 20-21 lines 424-461 to describe this as follows:

“**Multistep retrosynthesis prediction**

To verify the practical use in synthesis planning, we also extend our one-step model trained on the USPTO-50k dataset to full pathway design by sequentially performing retrosynthetic predictions. We choose 3 target compounds as examples, all of which

have significant medicinal importance, including the oral SARS-CoV-2 M^{pro} inhibitor Nirmatrelvir for treatment of COVID-19¹⁴, the third-generation EGFR inhibitor Osimertinib for treatment of non-small cell lung carcinoma¹⁵ and the Lenalidomide for treatment of multiple myeloma¹⁶. Note that none of these input structures (products and intermediates) in the three examples appears as a product in our training set. As shown in Fig. 10, our method successfully reproduces the complete synthetic pathway for these compounds.

The first example for Nirmatrelvir has been reported in the literature by Pfizer¹⁷ (Fig. 10a). Although the synthetic pathway consists of six reaction steps, our method succeeds at the rank-1 prediction for all steps except the third one predicted at rank-6, which directly demonstrates the superiority of our method. The first and second steps, which are the core reactions, can be easily reproduced by our model as dehydration of the amide to form the cyano group, followed by a condensation reaction to yield the key intermediate. The subsequent step is an amine ester exchange reaction, preceded by the common deprotection and ester hydrolysis, and the final step involves the amide formation, which exactly matches the published synthesis. The second example is the retrosynthetic pathway planning of Osimertinib, as depicted in Fig. 10b. Finlay et al.¹⁸ proposed a five-step reaction pathway for this drug, which is derived from readily available starting materials. Our model first suggests an acylation reaction with acryloyl chloride and then correctly predicts a reduction of the nitro group with rank-1. In the next two steps, sequential nucleophilic aromatic substitution reactions (S_NAr) are predicted to introduce amino side chain and nitroaniline. And the final step, unlike the Friedel–Crafts arylation reported in the literature, our model suggests a Suzuki cross-coupling reaction to produce 3-pyrazinyl indole. In the third example, the retrosynthesis pathway planning for Lenalidomide has also been demonstrated by Retrosim¹⁹ and LocalRetro²⁰ models, and our model can perfectly recover the route suggested by the Retrosim method. The first and third steps are suggested as the nitro reduction and the bromination with N-bromosuccinimide (NBS), which are also consistent with published literature pathway²¹. And in the second step, our model predicts a formation of the five-membered ring with the acid chloride, rather than the methyl ester, which is

feasible in synthetic chemistry. These results in three examples clearly show that our approach can generate nearly identical retrosynthetic pathways to those in the literature, mostly within the rank-2 predictions, and further demonstrate the great potential of our model for practical multistep retrosynthesis.

a. Nirmatrelvir

b. Osimertinib

c. Lenalidomide

Fig. 10 Multistep retrosynthesis predictions by Graph2Edits. **a** Nirmatrelvir, **b** Osimertinib and **c** Lenalidomide.

Reviewer 3

The authors present an approach for AI-based single-step retrosynthesis relying on a semi-template method. The approach is interesting even if it still relies on the knowledge of the templates for training. The most interesting part is the autoregressive prediction of the edits, which mimics the “reaction mechanism”, making the single-step retrosynthesis models more interpretable.

Reply: We thank the reviewer for his/her positive comments. In this revision, we have followed the suggestions made by the reviewer and updated the manuscript accordingly.

Reviewer Point P3.1 — The authors evaluate the model on USPTO 50k, which is quite a limited dataset. An evaluation on USPTO full would be more insightful.

Reply: We sincerely appreciate the reviewer for this valuable feedback that we have used to improve the quality of our manuscript. Admittedly, USPTO-50k is a small dataset with only 50016 reactions, but it is a clean dataset containing 10 reaction types commonly used by medicinal chemists and is widely used in previous retrosynthesis work⁷. To address this issue, we evaluated our model on the larger USPTO-full dataset to verify the scalability, and the performance result were shown in Table 3. Although the USPTO-full is much noisier than the clean USPTO-50k, our method, Graph2Edits, still achieved the top-1 accuracy of 44.0%, which is significantly higher than the baseline MEGAN, and is close to the state-of-the-art models. These results further demonstrate the scalability of the model. We have added Table 2 and two new paragraphs to the Section 2.1 and 2.2 on page 10-11 lines 213-216 and page 13 lines 264-271 in the revised manuscript.

“Additionally, we also use the original USPTO-full dataset from the entire USPTO (1976-Sep2016) to verify the scalability of our model. We use exactly the same splits as Dai et al.⁸, which contain approximately 800k/100k/100k training/validation/test reactions, and repeat the procedures given in the above USPTO-50k dataset processing.”

Table 3 Top- k exact match accuracy of the proposed Graph2Edits and baselines on USPTO-full dataset.

Category	Model	Top- k accuracy (%)			
		$k=1$	3	5	10
Template-Based	Retrosim	32.8	-	-	56.1
	Neuralsym	35.8	-	-	60.8
	GLN	39.3	-	-	63.7
Template-Free	Aug.Transformer	46.2	-	-	73.3
	GTA	46.6	-	-	70.4
	Graph2SMILES	45.7	-	-	63.4
Semi-Template-Based	RetroPrime	44.1	59.1	62.8	68.5
	MEGAN	33.6	-	-	63.9
	Graph2Edits (D-MPNN)	44.0	60.9	66.8	72.5

“We also compare the performance of Graph2Edits on the larger USPTO-full dataset with other baselines for retrosynthesis prediction. The results are presented in Table 3. Although the USPTO-full is much noisier than the clean USPTO-50k, our method still has competitive performance with a top-1 accuracy of 44.0%, on par with the semi-template-based method RetroPrime and outperforming MEGAN by a large margin. In addition, on larger k ($k > 1$), especially top-10 accuracy, Graph2Edits significantly outperforms all other methods except Aug.Transformer, showing similar superiority to the performance on the USPTO-50k dataset.”

Reviewer Point P3.2 — The metric used to evaluate the models is topk. While widely used in the AI-retrosynthesis community, is not the most appropriate metric for the problem. As the authors point out as well this is a one-to-many problem.

Reply: We thank the reviewer for this valuable comment. At present, the top-k exact match accuracy is the most commonly used evaluation metric assessing retrosynthetic model performance. However, as mentioned by the reviewers, evaluating retrosynthesis

models is challenging as multiple sets of reactants can be generated from the same product and the top-k exact match accuracy has serious limitations and underestimates model performance. It only compares the predicted reactions with those in the benchmark data, but does not consider novel predicted reactions that are not in the benchmark data but are highly likely. Such novel predictions should be assessed using existing data from very large reaction databases and evaluated from the perspective of a synthetic chemist, so to determine the accuracy and likelihood that these predicted approaches could be employed. These challenges are the future research directions of retrosynthesis prediction. Recently, there have a few alternative metrics been suggested to evaluate retrosynthesis models such as MaxFrag²² and Round-trip²³. To more comprehensively evaluate our model, we also calculated the maximal fragment accuracy (MaxFrag) and compared to the existing state-of-the-art methods, the results were shown in Table 2. We have added the related descriptions to the Section 2.2 on page 11 lines 224-228 and page12-13 lines 258-260 as follows:

“We additionally adopt the maximum fragment (MaxFrag) accuracy to evaluate the performance of our model. The MaxFrag accuracy, inspired by classical retrosynthesis, is to calculate the exact match of only the largest fragment to overcome the prediction limitation due to the existence of unclear reagent reactions in the dataset.”

“In addition, the results of MaxFrag accuracy of our model tested on USPTO-50k are shown in Table 2. Graph2Edits outperforms all baselines by a large margin and reaches 59.2% MaxFrag accuracy at top-1 predictions.”

Table 2 Top-*k* MaxFrag accuracy of the proposed Graph2Edits and baselines on USPTO-50k dataset.

Model	Top- k accuracy (%)				
	k = 1	3	5	10	50
Aug.Transformer	58.5	73.0	85.4	90.0	-
LocalRetro	57.8	82.1	89.7	95.0	98.4
MEGAN	54.2	75.7	83.1	89.2	95.1
Graph2Edits (MPNN)	56.8	80.3	87.5	92.8	95.4
Graph2Edits (D-MPNN)	59.2	80.1	86.1	91.3	93.1

Reviewer Point P3.3 — Pay attention to English in the manuscript (e.g. Line 119-125), and be more precise in the sentences (e.g. line 45 is not Coley that first proposed retrosynthesis – this is something that has been around for much longer, it was just not handled yet with AI.)

Reply: We sincerely thank the reviewer for careful reading and apologize there were problems with the English. We have tried our best to polish the language to improve the grammar and readability and the points mentioned by the reviewers have been revised as follows (line 41-43 and line 119-124):

“Retrosynthesis²⁴ is a method widely used by organic chemists to design synthetic routes to a target molecule by recursively decomposing it into simpler precursors.”

“As shown in Fig. 2, a simplified mechanism example in the Mitsunobu reaction: the reagent PPh₃ (triphenylphosphine) combines with DEAD (diethyl azodicarboxylate) to generate a phosphonium intermediate that binds to the alcohol oxygen, activating it as a leaving group, then the nucleophile oxygen anion and the phosphonium ion to perform nucleophilic substitution to yield the final product.”

Reviewer Point P3.4 — I believe more details are needed on how the edits are extracted from the atom mapping (perhaps a figure?). Moreover, I believe that not for all reactions these can be identified with the reaction mechanism. In many (some?) cases the edits are not the true reaction mechanism, but just an artifact.

Reply: We thank the reviewer for this valuable suggestion. We have added a new section ‘Details of graph edits’ to Supplementary Information of our manuscript as follows:

“Our graph edits are derived from the training set and represent the process of graph transformations in the retro-reactions. Since each atom is mapped on product and reactants, we mark the edit atoms or bonds to specify the positions and changes in each reaction. There are four different types of edits in reactions: (1) Delete bond, (2) Change bond, (3) Change atom, (4) Attach leaving group (LG) on atom, and our priority order for graph edits is Delete bond > Change bond > Change atom > Attach LG.”

a. Atom center reaction

b. Bond center reaction

c. Multiple centers reaction

Fig. 1 The examples of derived edits sequence in retro-reactions. **a** Atom center reaction, **b** Bond center reaction and **c** Multiple centers reaction. The two numbers in brackets of atom edit represent the number of hydrogen and the chiral type to be changed, and the two numbers in brackets of bond edit indicate the bond type and the bond stereo configuration to be changed. The changes in reaction are highlighted in red.

“As shown in Fig. 1a, the first edit is ('Change Atom', (0, 0)) on the atom 1, and the two numbers in brackets represent the number of hydrogen and the chiral type to be changed, respectively. And then, the graph edits is ('Attaching LG', '*C(=O)c1ccccc1C(*)=O') represents the '*C(=O)c1ccccc1C(*)=O' is added on the atom 1. At the reaction shown in Fig. 1b, the bond [2, 3] is deleted and then the '*N(C)C' and '*Br' group are added on the atom 2 and 3, respectively. At the bottom of Fig. 1c,

the edit is first to delete the bond [6, 7] and [10, 11], this sequence may not match the true reaction mechanism, but it does not affect the final result of the graph transformation. Next, the bond edits ('Change Bond', (2, 0)), ('Change Bond', (1, 0)), ('Change Bond', (1, 0)) are operated on bond [7, 8], [8, 10], [6, 11], and the two numbers in brackets of bond edit represent the bond type and the bond stereo configuration to be changed. Finally, the leaving group '*=O' and '*Br' are attached on the atom 11 and 6, respectively.”

a. Amino protection reaction

b. Aldehyde deprotection reaction

c. Boric acid esterification

Supplementary Figure 1. Examples of incorrect graph edits sequence derived from retro-reactions.

There are also some incorrect graph edits sequence which derived from a small number of reactions using our automatic preprocessing method. The examples are shown in Supplementary Fig. 1. A common feature of these reactions is that the same leaving group needs to be added to more than one atom. And since there is no bond addition in the predefined edits, our method cannot handle this. Fortunately, there is little reactions of new bond formation in the training set (about 0.1%)²⁵.

After generating the ground truth edits sequence based on the atom mapping in reactions, we build the edits vocabulary E . All graph edits were derived from the training set of USPTO-50k dataset, including 6 bond edits, 152 atom edits (7 Change Atom and 145 Attach LG), and a termination symbol and the details can be seen in Supplementary Table 1. The same procedure was used to build the edits vocabulary on USPTO-full dataset and the difference is that the edits Attach LG must appear at least 50 times in the training set of USPTO-full before it will be collected into the vocabulary E . this edits vocabulary E include 6 bond edits, 336 atom edits (8 Change Atom and 328 Attach LG), and a termination symbol.

Supplementary Table 1. Graph edits found on the USPTO-50k training set.

Number	Graph edits
1	('Delete Bond', (None, None))
2	('Change Bond', (2, 0))
3	('Change Bond', (1, 0))
4	('Change Bond', (3, 0))
5	('Change Bond', (2, 3))
6	('Change Bond', (2, 2))
7	('Change Atom', (1, 0))
8	('Change Atom', (0, 0))
9	('Change Atom', (1, 1))
10	('Change Atom', (1, 2))
11	('Change Atom', (0, 2))
12	('Change Atom', (2, 0))
13	('Change Atom', (0, 1))
14	('Attaching LG', '*C(=O)OCc1ccccc1')
15	('Attaching LG', '*O')
16	('Attaching LG', '*=O')
17	('Attaching LG', '*Cl')
18	('Attaching LG', '*Br')
19	('Attaching LG', '*B(O)O')
20	('Attaching LG', '*B1OC(C)(C)C(C)(C)O1')
21	('Attaching LG', '*C(C)(C)C')
22	('Attaching LG', '*Cc1ccc(OC)cc1')
23	('Attaching LG', '*C')
24	('Attaching LG', '*F')
25	('Attaching LG', '*Cc1ccccc1')
26	('Attaching LG', '*[Si](C)(C)C')
27	('Attaching LG', '*=P(c1ccccc1)(c1ccccc1)c1ccccc1')

28	('Attaching LG', '*OC(=O)OC(C)(C)C')
29	('Attaching LG', '*I')
30	('Attaching LG', '*[O-]')
31	('Attaching LG', '*OCC')
32	('Attaching LG', '*N1C(=O)CCC1=O')
33	('Attaching LG', '*OS(=O)(=O)C(F)(F)F')
34	('Attaching LG', '*OCC(F)(F)F')
35	('Attaching LG', '*CC')
36	('Attaching LG', '*OC')
37	('Attaching LG', '*C(=O)OC(C)(C)C')
38	('Attaching LG', '*N')
39	('Attaching LG', '*Cl')
40	('Attaching LG', '*[Si](C)(C)C(C)(C)C')
41	('Attaching LG', '*OC(C)=O')
42	('Attaching LG', '*C(C)=O')
43	('Attaching LG', '*[P+](c1ccccc1)(c1ccccc1)c1ccccc1')
44	('Attaching LG', '*[Sn](CCCC)(CCCC)CCCC')
45	('Attaching LG', '*N(C)C')
46	('Attaching LG', '*C(=O)OC')
47	('Attaching LG', '*OS(C)(=O)=O')
48	('Attaching LG', '*B1OCC(C)(C)CO1')
49	('Attaching LG', '*[Mg+]
50	('Attaching LG', '*C(=O)c1ccccc1C(*)=O')
51	('Attaching LG', '*OC(=O)c1ccccc1Cl')
52	('Attaching LG', '*OC(C)(C)C')
53	('Attaching LG', *OC(=O)CCCCCCCCCCCCCCCCCCCC')
54	('Attaching LG', '*P(=O)(OCC)OCC')
55	('Attaching LG', '*C(Cl)(Cl)Cl')
56	('Attaching LG', '*OC(=O)C(F)(F)F')
57	('Attaching LG', '*OCC(C)C')
58	('Attaching LG', '*[Cu]
59	('Attaching LG', '*OS(=O)O')
60	('Attaching LG', '*OC(=O)CC')
61	('Attaching LG', '*B1OC(C)C(C)O1')
62	('Attaching LG', '*N(C)OC')
63	('Attaching LG', '*[Zn+]
64	('Attaching LG', '*C(Br)(Br)Br')
65	('Attaching LG', '*C(=O)CCl')
66	('Attaching LG', '*OC=O')
67	('Attaching LG', '*C1CCCCO1')
68	('Attaching LG', '*=[N+]=[N-]')
69	('Attaching LG', '*C(=O)C(C)C')

70	('Attaching LG', '*[S+](C)(C)=O')
71	('Attaching LG', '*=C')
72	('Attaching LG', '*C(=O)CCCCCCCCCCCCC')
73	('Attaching LG', '*OC(=O)OCC')
74	('Attaching LG', '*C(Br)Br')
75	('Attaching LG', '*OCC(Cl)(Cl)Cl')
76	('Attaching LG', '*C(=O)C(F)(F)F')
77	('Attaching LG', '*[Sn](CC)(CC)CC')
78	('Attaching LG', '*[S+](C)C')
79	('Attaching LG', '*C(=O)CCC')
80	('Attaching LG', '*[Si](C(C)C)(C(C)C)C(C)C')
81	('Attaching LG', '*COC')
82	('Attaching LG', '*[Sn](C)(C)C')
83	('Attaching LG', '*C(=O)c1cccc1')
84	('Attaching LG', '*C(=O)OCC')
85	('Attaching LG', '*OCCC')
86	('Attaching LG', '*OC(=O)C(C)(C)C')
87	('Attaching LG', '*C(=O)OCC1c2cccc2-c2cccc21')
88	('Attaching LG', '*OC(C)Cl')
89	('Attaching LG', '*[P+](C)(C)C')
90	('Attaching LG', '*OCCCCCCCCCCCC')
91	('Attaching LG', '*n1ccnc1')
92	('Attaching LG', '*B1OB(C)OB(C)O1')
93	('Attaching LG', '*B1OCCN(c2cccc2)CCO1')
94	('Attaching LG', '*B1OC(=O)CN(C)CC(=O)O1')
95	('Attaching LG', '*B1OB(C=C)OB(C=C)O1')
96	('Attaching LG', '*C(=O)CC')
97	('Attaching LG', '*C(c1cccc1)c1cccc1')
98	('Attaching LG', '*OC(=O)CCCCCCCC')
99	('Attaching LG', '*OC(=O)CCI')
100	('Attaching LG', '*OC(=O)C(F)(F)Cl')
101	('Attaching LG', '*OC(=O)OCc1cccc1')
102	('Attaching LG', '*P(=O)(OC)OC')
103	('Attaching LG', '*OC(=O)C(C)C')
104	('Attaching LG', '*C(=O)c1ccc([N+](=O)[O-])cc1')
105	('Attaching LG', '*OC(Cl)(Cl)Cl')
106	('Attaching LG', '*[Zn]Br')
107	('Attaching LG', '*[Si](C)(C)C(C)(C)C(C)C')
108	('Attaching LG', '*OCc1cccc1')
109	('Attaching LG', '*B1OC(C)CC(C)(C)O1')
110	('Attaching LG', '*B(OC(C)C)OC(C)C')
111	('Attaching LG', '*OCCCC')
112	('Attaching LG', '*OC(C)C(C)=O')

113	('Attaching LG', '*=N[Si](C)(C)C')
114	('Attaching LG', '*OC1NCC(=C)O1')
115	('Attaching LG', '*=S')
116	('Attaching LG', '*OC(=O)c1ccccc1Cl')
117	('Attaching LG', '*C(=O)N(C)C')
118	('Attaching LG', '*=C(Cl)CCCC')
119	('Attaching LG', '*Cc1cc([N+](=O)[O-])ccc1O')
120	('Attaching LG', '*[Si](CC)(CC)CC')
121	('Attaching LG', '*B1OCCO1')
122	('Attaching LG', '*OC(=O)c1ccccc1')
123	('Attaching LG', '*[Mg]Br')
124	('Attaching LG', '*OC(=O)C(F)F')
125	('Attaching LG', '*CCCC')
126	('Attaching LG', '*OC(=O)CCCC')
127	('Attaching LG', '*C(C)C')
128	('Attaching LG', '*C(=O)OCC=C')
129	('Attaching LG', '*C(=O)[C@H](C)c1ccc2cc(OC)ccc2c1')
130	('Attaching LG', '*C(=O)C(C)(C)C')
131	('Attaching LG', '*OC(=O)C(=C)C')
132	('Attaching LG', '*[N+](C)(C)C')
133	('Attaching LG', '*C(=O)[C@@H](OC(C)=O)c1ccccc1')
134	('Attaching LG', '*[Zn]I')
135	('Attaching LG', '*S(C)(=O)=O')
136	('Attaching LG', '*OC(=O)CCCC')
137	('Attaching LG', '*B1OC=CC(C(=O)O)O1')
138	('Attaching LG', '*OC(=O)C1CC1')
139	('Attaching LG', '*OCC(C)(C)NS(=O)(=O)C(F)(F)F')
140	('Attaching LG', '*OC(=O)CBr')
141	('Attaching LG', '*C(=O)/C=C/C=C/C=C/C=C/C(=O)O')
142	('Attaching LG', '*B1OB(c2ccc(F)cc2)OB(c2ccc(F)cc2)O1')
143	('Attaching LG', '*OC(=O)CCC')
144	('Attaching LG', '*OC(=O)C(=NOC)c1csc(NC(c2ccccc2)(c2ccccc2)c2ccccc2)n1')
145	('Attaching LG', '*C(=O)c1ccc(OCCCCC)cc1')
146	('Attaching LG', '*OC(=O)/C=C\\n1cnc(-c2cc(OC)cc(C(F)(F)F)c2)n1')
147	('Attaching LG', '*C(=O)c1ccc(OC)cc1')
148	('Attaching LG', '*P(=O)(OCC(F)(F)F)OCC(F)(F)F')
149	('Attaching LG', '*C(=O)CCCC')
150	('Attaching LG', '*OC(=O)Cl')
151	('Attaching LG', '*=NS(=O)(=O)C(F)(F)F')
152	('Attaching LG', '*OCc1ccc([N+](=O)[O-])cc1')

153	('Attaching LG', '*C(=O)C1(C)CCCCC1')
154	('Attaching LG', '*C(=O)CCCCBr')
155	('Attaching LG', '*C(=O)CCC(=O)OC')
156	('Attaching LG', '*C(=O)c1cccc(Cl)c1')
157	('Attaching LG', '*B1OCCCCO1')
158	('Attaching LG', '*OC(C)C')
159	'Terminate'

Reviewer Point P3.5 — The authors claim that their model improves on rare reactions. However, with such a small dataset, split randomly, is hard to make such a claim. The authors would first need to check for the same rare reactions they extract, how the state-of-the-art performs and possibly also use a better split (ex. Tanimoto-based splitting).

Reply: We appreciate the reviewer for helpful comments. In the Section 2.3, we evaluated the performance of our model on reactions with the different edits sequence length and found that with the increase of edits sequence length, the number of reaction samples decreased dramatically, while the performance of our model did not decrease significantly. Thus, we concluded that our model remained relatively robust in rare reactions. Following the reviewer’s suggestion, we further investigated the performance on reactions with different reaction similarities which based on Tanimoto coefficient, and we have added a new paragraph to the Section 2.3 on page 15 lines 308-320 as follows:

“To further investigate the performance on reactions with different reaction similarities, we use Tanimoto coefficient⁹ to calculate the reaction similarities. For each reaction, we calculate the 2048-bit reaction fingerprints¹⁰, and the similarity is quantified by the mean Tanimoto similarity between the reaction and all other reactions in the test set. We cluster the reactions according to their reaction similarity using the K-means clustering algorithm in Euclidean distances¹¹. Table 4 presents the clustering results, and we can see that the clusters with high reaction similarity generally have higher top-*k* accuracy. It is worth noting that cluster 1, which has the lowest similarity

with other reactions, can be regarded as a rare reaction cluster, and its top-1 accuracy can also reach nearly 50%. These results demonstrate that our performance is not obtained by overfitting to one particular category of reactions. Such a trait is also crucial since retrosynthesis may involve rare reactions that haven't been thoroughly explored in the literature.”

Table 4 Top-*k* exact match accuracy per each cluster with different reaction similarity.

Cluster	Cluster size	Mean similarity	Top- k accuracy (%)			
			k = 1	3	5	10
1	181	0.019	49.2	69.1	73.5	82.3
2	326	0.032	56.4	76.7	82.8	87.1
3	517	0.042	56.1	78.3	85.5	90.7
4	631	0.050	51.2	75.1	81.3	89.1
5	663	0.057	52.6	75.0	83.3	89.3
6	684	0.065	56.0	78.2	83.5	90.2
7	702	0.072	56.8	77.5	84.3	89.6
8	605	0.079	56.9	78.1	84.1	89.1
9	448	0.086	57.4	80.1	85.5	91.3
10	247	0.096	56.7	80.2	84.6	90.3

Reviewer Point P3.6 — The authors claim that template-free approaches output many invalid predictions. However, I guess also their model can generate “invalid edits” that do not bring to a valid chemical structure. Can this be quantified? And possibly compared to the topk percentage of invalid SMILES generated by state-of-the-art template-free approaches?

Reply: We thank the reviewer for this valuable suggestion. To evaluate the invalid rates, we calculated the top-k percentage of invalid smiles generated by Graph2Edits and compared it with the template-free models seq2seq and SCROP. We have added a new

section ‘Evaluation of invalid rates’ to Supplementary Information of our manuscript as follows:

“The template-free approaches can output grammatically invalid predictions and our model Graph2Edits can also generate “invalid edits” that do not bring to a valid chemical structure. To evaluate the invalid rates, we calculated the top- k percentage of invalid smiles generated by Graph2Edits and compared it with the template-free models seq2seq⁷ and SCROP²⁶. As shown in Supplementary Table 4, the top-1 predictions generated by our model are all valid chemical structures, and only 0.71% and 0.87% of the the top-10 predictions are grammatically invalid, which is significantly better than those by the template-free models seq2seq and SCROP (used the syntax corrector).”

Supplementary Table 4. Comparison of invalid rates among seq2seq, SCROP, and our model Graph2Edits for different beam sizes on USPTO-50k dataset.

Model	Top- k invalid rate (%)			
	$k = 1$	3	5	10
seq2seq	12.2	15.3	18.4	22.0
SCROP	0.7	1.4	1.8	2.3
Graph2Edits	0	0.71	0.86	0.87

References

- 1 Herges, R. Organizing principle of complex reactions and theory of coarctate transition states. *Angewandte Chemie International Edition in English* **33**, 255-276 (1994).
- 2 Bradshaw, J., Kusner, M., Paige, B., Segler, M. & Hernández-Lobato, J. A generative model for electron paths. In: *International Conference on Learning Representations*. (2019).
- 3 Fooshee, D. et al. Deep learning for chemical reaction prediction. *Molecular Systems Design & Engineering* **3**, 442-452 (2018).
- 4 Do, K., Tran, T. & Venkatesh, S. Graph transformation policy network for chemical reaction prediction. In: *International Conference on Knowledge Discovery & Data Mining*. 750-760 (2019).

- 5 Bi, H. et al. Non-Autoregressive Electron Redistribution Modeling for Reaction Prediction. In: *International Conference on Machine Learning*. 904-913 (2021).
- 6 Sacha, M. et al. Molecule edit graph attention network: modeling chemical reactions as sequences of graph edits. *Journal of Chemical Information and Modeling* **61**, 3273-3284 (2021).
- 7 Liu, B. et al. Retrosynthetic reaction prediction using neural sequence-to-sequence models. *ACS central science* **3**, 1103-1113 (2017).
- 8 Dai, H., Li, C., Coley, C., Dai, B. & Song, L. Retrosynthesis prediction with conditional graph logic network. *Advances in Neural Information Processing Systems* **32**, 8872-8882 (2019).
- 9 Bajusz, D., Rácz, A. & Héberger, K. Why is Tanimoto index an appropriate choice for fingerprint-based similarity calculations? *Journal of cheminformatics* **7**, 1-13 (2015).
- 10 Schneider, N., Lowe, D. M., Sayle, R. A. & Landrum, G. A. Development of a novel fingerprint for chemical reactions and its application to large-scale reaction classification and similarity. *Journal of chemical information and modeling* **55**, 39-53 (2015).
- 11 Likas, A., Vlassis, N. & Verbeek, J. J. The global k-means clustering algorithm. *Pattern recognition* **36**, 451-461 (2003).
- 12 Chen, Z., Ayinde, O. R., Fuchs, J. R., Sun, H. & Ning, X. G²Retro: Two-Step Graph Generative Models for Retrosynthesis Prediction. Preprint at <https://arxiv.org/abs/2206.04882> (2022).
- 13 Jin, W., Coley, C., Barzilay, R. & Jaakkola, T. Predicting organic reaction outcomes with weisfeiler-lehman network. *Advances in neural information processing systems* **30**, 2604-2613 (2017).
- 14 Hammond, J. et al. Oral nirmatrelvir for high-risk, nonhospitalized adults with Covid-19. *New England Journal of Medicine* **386**, 1397-1408 (2022).
- 15 Greig, S. L. Osimertinib: first global approval. *Drugs* **76**, 263-273 (2016).
- 16 Palumbo, A. et al. Continuous lenalidomide treatment for newly diagnosed multiple myeloma. *New England Journal of Medicine* **366**, 1759-1769 (2012).
- 17 Owen, D. R. et al. An oral SARS-CoV-2 Mpro inhibitor clinical candidate for the treatment of COVID-19. *Science* **374**, 1586-1593 (2021).
- 18 Finlay, M. R. V. et al. Discovery of a potent and selective EGFR inhibitor (AZD9291) of both sensitizing and T790M resistance mutations that spares the wild type form of the receptor. *Journal of Medicinal Chemistry* **57**, 8249-8267 (2014).
- 19 Coley, C. W., Rogers, L., Green, W. H. & Jensen, K. F. Computer-assisted retrosynthesis based on molecular similarity. *ACS central science* **3**, 1237-1245 (2017).
- 20 Chen, S. & Jung, Y. Deep retrosynthetic reaction prediction using local reactivity and global attention. *JACS Au* **1**, 1612-1620 (2021).
- 21 Ponomaryov, Y. et al. Scalable and green process for the synthesis of anticancer drug lenalidomide. *Chemistry of Heterocyclic Compounds* **51**, 133-138 (2015).

- 22 Tetko, I. V., Karpov, P., Van Deursen, R. & Godin, G. State-of-the-art augmented NLP transformer models for direct and single-step retrosynthesis. *Nature communications* **11**, 1-11 (2020).
- 23 Schwaller, P. et al. Predicting retrosynthetic pathways using transformer-based models and a hyper-graph exploration strategy. *Chemical science* **11**, 3316-3325 (2020).
- 24 Corey, E. J. The logic of chemical synthesis: multistep synthesis of complex carbogenic molecules (nobel lecture). *Angewandte Chemie International Edition in English* **30**, 455-465 (1991).
- 25 Somnath, V. R., Bunne, C., Coley, C., Krause, A. & Barzilay, R. Learning graph models for retrosynthesis prediction. *Advances in Neural Information Processing Systems* **34**, 9405-9415 (2021).
- 26 Zheng, S., Rao, J., Zhang, Z., Xu, J. & Yang, Y. Predicting retrosynthetic reactions using self-corrected transformer neural networks. *Journal of chemical information and modeling* **60**, 47-55 (2019).

REVIEWER COMMENTS

Reviewer #1 (Remarks to the Author):

I have reread the manuscript by Chen et al. subjected to my revision. To be honest, I found less doubtful statements in the previous iteration. In particular, the authors keep drawing conclusions from the TopK metric. Although this is not the best metric for the retrosynthesis problem, it is widely used to compare models, so its utilization in this work is justified. Nevertheless, this metric measures how similar is the output reaction to the one found in USPTO, not how good the chemistry proposed by the model is. In particular, the reaction given in USPTO can have a very small yield, which could be optimized 10x if another method was used. Or sometimes the reactions are just wrong. In other words, USPTO should not in fact be regarded as a ground truth for chemistry.

As long as the authors treat TopK on USPTO as a way to compare to other models it is fine. But the authors make an additional step. They try to interpret the non-USPTO results as errors. In particular, in section 2.3, in the discussion of reactions in Fig. S2b, the authors write "the model identifies both the bond edit and leaving group correctly on top-2 prediction, but fails to predict the two chiral changes." In fact, this is a perfect example of problems in USPTO - the top2 prediction is perfectly fine, the "ground-truth" prediction in USPTO is probably wrong. I don't see any easy way of introducing stereochemistry far from the reaction center, as it is visible in the "ground-truth" reaction, and therefore the correct reaction should start from chiral substrates, as in Top2 prediction. Instead of making such an observation, the authors try to justify the "error" done by the model. Similarly, there is nothing wrong with substituting I for Cl in Fig. S2a. Usually, in organic chemistry all halides (Cl, Br, I) are interchangeable.

If the authors want to have the analysis of correct/incorrect predictions (which I think they should), they should ask professional organic chemists to assess some random reactions, actually in the same way it was done in MEGAN paper.

When talking about MEGAN, after rereading the manuscript I cannot see what is the key difference which makes Graph2Edits a better model than MEGAN. As far as I understand, the underlying idea is the same - propose consecutive graph edits to obtain substrates for a given product. MEGAN, however, is very creative - it proposes many "magic" reactions, where some chemically impossible connections of carbon atoms take place. If the authors could show, that their model is better in this aspect (so in fact make better error analysis), that would be an important input.

Also, in the line of comparing to USPTO, the authors make an interesting analysis of similar reactions (section 2.4). However, the similarity measure is based on Tanimoto coefficient. First, I am not sure, how the authors use it on reactions, as the coefficient is calculated for compounds. But more importantly, the Tanimoto coefficient, which is based on some local similarities, may give similar values for chemically distinct compounds (take e.g. heterocyclic compounds), or vice-versa - distinct values for chemically similar structures (where chloride is substituted to iodine, etc.). I doubt if it can efficiently classify reactions. In fact, such classification is usually done based on templates. Did the authors check, that their analysis based on Tanimoto coefficient indeed separates chemically different reactions? For example, are all electrophilic aromatic substitutions in one class? And electrophilic aromatic substitutions in a separate class? This also applies to the whole section 2.6, where the similarities between reactions are analyzed. Are the clusters really showing different reactions?

Also, the authors identify the reactions, for which the model proposes more edits with "rare" reactions (line 305). The counterexample is all the heterocycles synthesis - all of them are rather complicated but are really frequent in USPTO (but in USPTO-50k they comprise only 1.8%!).

Also, the discussion of stereochemistry (section 2.4, lines 326+) rises my concerns. After removing the stereochemistry information, "the model has deeply learned the transformation rules of these reactions...". But "these reactions" are just ordinary reactions, for which the stereochemistry information was included. So it would be rather odd if the model could not learn those reactions. Also, with the stereochemistry the "performance drops", but the "top-10 accuracy still achieves 66.2%".

Based on the previous discussion of TopK on USPTO, I don't think this argument is convincing. I think, that again, when judging the model efficiency on reactions with stereochemistry, one should check them one by one. The authors finalize the section with "it has an advantage in predicting stereochemical reactions...". There is no such term as "stereochemical reaction". The reaction can be stereospecific, or stereoselective. But I think in this case the authors mean just a reaction, for which the stereo-information is given in USPTO, which is rather random. So I would not draw conclusions here.

There are also some less critical statements I am not sure about. Line 89 - "Transformers have a lower [compared to template-based methods] computational cost". I am not sure if that is true - transformers are known to be efficient, but rather slow in production, don't they?

Also, I think that one of the substrates in Fig. 6c should not be aromatic, and the reaction in Fig 7c is in fact dehydration.

Finally, I found some typos and grammar problems:

- line 158 - adding A new atom
- line 162 - Rather THAN only
- line 171 - AND is unnecessary
- line 176 - and have been -> WHICH have been
- line 181 - "same" is not needed
- line 279 - leaving group IS
- line 286 - of THE model
- line 357 - Evaluation OF the
- line 456 - unnecessary "in three examples"
- line 457 - pathways AS those
- line 515 - missing whitespace after "where"
- line 544 - unnecessary "the"
- caption to Fig. 5 - there should be violet instead of blue

To sum up, I like the idea of the paper, and I like the analysis and the examples, but I think right now the authors are going too far with their conclusions, which stems from a bit careless usage of tools and metrics in my opinion.

Reviewer #2 (Remarks to the Author):

I have one further comment on the accuracy of the model as described in the paper. At the second point on the main contribution, the authors state that D-MPNN can improve the predictive performance. If one sees Table 1, however, the prediction accuracy is higher at top-1 and top-3 (marginally) only but lower at top-5, 10, 50 compared to MPNN-based model. Since retrosynthesis is potentially a one-to-many prediction, higher accuracy at high k can be quite important. Also, can the authors provide some reasoning why higher k accuracy is lower, and only top-1/3 accuracy is improved?

Reviewer #3 (Remarks to the Author):

In general I am satisfied with the modifications to the manuscript and with the answers the authors provided. Two things I would like to point out:

1) MaxFrag is not much more relevant than topk. I would encourage the authors to use Round-Trip accuracy. While not perfect, can be easily evaluated by training a Transformer model for reaction prediction on USPTO-50k

2) The analysis with Tanimoto similarity is not exactly what I intended to see, because the training set used to train the model seems to be the same from the random splitting. The idea was to generate a new train/validation/test split maximising the diversity between the sets. Similar to this paper: <https://www.nature.com/articles/s41467-021-21895-w>

REVIEWER COMMENTS

Reviewer #1 (Remarks to the Author):

I have reread the manuscript by Chen et al. subjected to my revision. To be honest, I found less doubtful statements in the previous iteration. In particular, the authors keep drawing conclusions from the TopK metric. Although this is not the best metric for the retrosynthesis problem, it is widely used to compare models, so its utilization in this work is justified. Nevertheless, this metric measures how similar is the output reaction to the one found in USPTO, not how good the chemistry proposed by the model is. In particular, the reaction given in USPTO can have a very small yield, which could be optimized 10x if another method was used. Or sometimes the reactions are just wrong. In other words, USPTO should not in fact be regarded as a ground truth for chemistry.

As long as the authors treat TopK on USPTO as a way to compare to other models it is fine. But the authors make an additional step. They try to interpret the non-USPTO results as errors. In particular, in section 2.3, in the discussion of reactions in Fig. S2b, the authors write "the model identifies both the bond edit and leaving group correctly on top-2 prediction, but fails to predict the two chiral changes." In fact, this is a perfect example of problems in USPTO - the top2 prediction is perfectly fine, the "ground-truth" prediction in USPTO is probably wrong. I don't see any easy way of introducing stereochemistry far from the reaction center, as it is visible in the "ground-truth" reaction, and therefore the correct reaction should start from chiral substrates, as in Top2 prediction. Instead of making such an observation, the authors try to justify the "error" done by the model. Similarly, there is nothing wrong with substituting I for Cl in Fig. S2a. Usually, in organic chemistry all halides (Cl, Br, I) are interchangeable. If the authors want to have the analysis of correct/incorrect predictions (which I think they should), they should ask professional organic chemists to assess some random reactions, actually in the same way it was done in MEGAN paper.

When talking about MEGAN, after rereading the manuscript I cannot see what is the key difference which makes Graph2Edits a better model than MEGAN. As far as I understand, the underlying idea is the same - propose consecutive graph edits to obtain substrates for a given product. MEGAN, however, is very creative - it proposes many "magic" reactions, where some chemically impossible connections of carbon atoms take place. If the authors could show, that their model is better in this aspect (so in fact make better error analysis), that would be an important input.

Also, in the line of comparing to USPTO, the authors make an interesting analysis of similar reactions (section 2.4). However, the similarity measure is based on Tanimoto coefficient. First, I am not sure, how the authors use it on reactions, as the coefficient is calculated for compounds. But more importantly, the Tanimoto coefficient, which is based on some local similarities, may give similar values for chemically distinct compounds (take e.g. heterocyclic compounds), or vice-versa - distinct values for chemically similar structures (where chloride is substituted to iodine, etc.). I doubt if it can efficiently classify reactions. In fact, such classification is usually done based on templates. Did the authors check, that their analysis based on Tanimoto coefficient indeed separates chemically different reactions? For example, are all electrophilic aromatic substitutions in one class? And electrophilic aromatic substitutions in a separate class? This also applies to the whole section 2.6, where the similarities between reactions are analyzed. Are the clusters really showing different reactions?

Also, the authors identify the reactions, for which the model proposes more edits with "rare" reactions (line 305). The counterexample is all the heterocycles synthesis - all of them are rather complicated but are really frequent in USPTO (but in USPTO-50k they comprise only 1.8%!).

Also, the discussion of stereochemistry (section 2.4, lines 326+) rises my concerns. After removing the stereochemistry information, "the model has deeply learned the transformation rules of these reactions...". But "these reactions" are just ordinary

reactions, for which the stereochemistry information was included. So it would be rather odd if the model could not learn those reactions. Also, with the stereochemistry the "performance drops", but the "top-10 accuracy still achieves 66.2%". Based on the previous discussion of TopK on USPTO, I don't think this argument is convincing. I think, that again, when judging the model efficiency on reactions with stereochemistry, one should check them one by one. The authors finalize the section with "it has an advantage in predicting stereochemical reactions...". There is no such term as "stereochemical reaction". The reaction can be stereospecific, or stereoselective. But I think in this case the authors mean just a reaction, for which the stereo-information is given in USPTO, which is rather random. So I would not draw conclusions here.

There are also some less critical statements I am not sure about. Line 89 - "Transformers have a lower [compared to template-based methods] computational cost". I am not sure if that is true - transformers are known to be efficient, but rather slow in production, don't they?

Also, I think that one of the substrates in Fig. 6c should not be aromatic, and the reaction in Fig 7c is in fact dehydration.

Finally, I found some typos and grammar problems:

- line 158 - adding A new atom
- line 162 - Rather THAN only
- line 171 - AND is unnecessary
- line 176 - and have been -> WHICH have been
- line 181 - "same" is not needed
- line 279 - leaving group IS
- line 286 - of THE model
- line 357 - Evaluation OF the
- line 456 - unnecessary "in three examples"
- line 457 - pathways AS those

- line 515 - missing whitespace after "where"
- line 544 - unnecessary "the"
- caption to Fig. 5 - there should be violet instead of blue

To sum up, I like the idea of the paper, and I like the analysis and the examples, but I think right now the authors are going too far with their conclusions, which stems from a bit careless usage of tools and metrics in my opinion.

Reviewer #2 (Remarks to the Author):

I have one further comment on the accuracy of the model as described in the paper. At the second point on the main contribution, the authors state that D-MPNN can improve the predictive performance. If one sees Table 1, however, the prediction accuracy is higher at top-1 and top-3 (marginally) only but lower at top-5, 10, 50 compared to MPNN-based model. Since retrosynthesis is potentially a one-to-many prediction, higher accuracy at high k can be quite important. Also, can the authors provide some reasoning why higher k accuracy is lower, and only top-1/3 accuracy is improved?

Reviewer #3 (Remarks to the Author):

In general I am satisfied with the modifications to the manuscript and with the answers the authors provided. Two things I would like to point out:

1) MaxFrag is not much more relevant than topk. I would encourage the authors to use Round-Trip accuracy. While not perfect, can be easily evaluated by training a Transformer model for reaction prediction on USPTO-50k

2) The analysis with Tanimoto similarity is not exactly what I intended to see, because the training set used to train the model seems to be the same from the random splitting. The idea was to generate a new train/validation/test split maximising the diversity between the sets. Similar to this paper: <https://www.nature.com/articles/s41467-021-21895-w>

Reviewer 1

I have reread the manuscript by Chen et al. subjected to my revision. To be honest, I found less doubtful statements in the previous iteration. In particular, the authors keep drawing conclusions from the TopK metric. Although this is not the best metric for the retrosynthesis problem, it is widely used to compare models, so its utilization in this work is justified. Nevertheless, this metric measures how similar is the output reaction to the one found in USPTO, not how good the chemistry proposed by the model is. In particular, the reaction given in USPTO can have a very small yield, which could be optimized 10x if another method was used. Or sometimes the reactions are just wrong. In other words, USPTO should not in fact be regarded as a ground truth for chemistry.

Reply: We sincerely thank the reviewer for the thorough review, constructive comments, and very useful suggestions. It is very helpful to improve the quality of our manuscript. Current retrosynthesis work is indeed lack of reasonable evaluation metrics and evaluating retrosynthesis models is challenging as there are potentially multiple valid ways to synthesize a product. The top-k exact match accuracy, which is to evaluate the ability of the model to reproduce the results recorded in the dataset, is the most commonly used metric in the current retrosynthesis prediction research. The use of top-k accuracy as the evaluation metric to compare the performance of models is a good benchmark and relatively fair and justified, although not perfect. Evaluation of how good the chemistry proposed by the retrosynthesis model usually requires synthetic chemists to verify the predicted results one by one and this may not be feasible. An alternative is to use a well-trained forward reaction prediction model as a digital domain expert to replace human chemists for evaluating the correctness of the predictions generated by the retrosynthetic model and this evaluation metric is called round-trip accuracy¹. To more comprehensively evaluate our model, we have also calculated the round-trip accuracy and compared to the existing state-of-the-art semi-template-based methods, the results were shown in Table 2 and the top-1 round-trip

accuracy of our model reaches nearly 86%. These descriptions can be seen in Reviewer Point P 3.1.

On the dataset level, the USPTO dataset and its variants are the most popular datasets in academic research due to their open and easy accessibility. Although the USPTO dataset has the problems of inaccurate atom-mapping information, noisy stereochemical data, and even some wrong reactions¹⁻³, there is no better alternative dataset. A decent dataset for deep learning models is extremely important and future works on constructing suitable datasets and solving contradictory information from datasets are urgent and crucial.

Reviewer Point P1.1 — As long as the authors treat TopK on USPTO as a way to compare to other models it is fine. But the authors make an additional step. They try to interpret the non-USPTO results as errors. In particular, in section 2.3, in the discussion of reactions in Fig. S2b, the authors write "the model identifies both the bond edit and leaving group correctly on top-2 prediction, but fails to predict the two chiral changes." In fact, this is a perfect example of problems in USPTO - the top2 prediction is perfectly fine, the "ground-truth" prediction in USPTO is probably wrong. I don't see any easy way of introducing stereochemistry far from the reaction center, as it is visible in the "ground-truth" reaction, and therefore the correct reaction should start from chiral substrates, as in Top2 prediction. Instead of making such an observation, the authors try to justify the "error" done by the model. Similarly, there is nothing wrong with substituting I for Cl in Fig. S2a. Usually, in organic chemistry all halides (Cl, Br, I) are interchangeable.

If the authors want to have the analysis of correct/incorrect predictions (which I think they should), they should ask professional organic chemists to assess some random reactions, actually in the same way it was done in MEGAN paper.

Reply: We thank the reviewer for this critical comment that indeed improved the quality of this manuscript. We do agree with the reviewer that we cannot treat the non-USPTO results as errors, and the predictions differ from the ground-truth reactants are

usually feasible and can be executed using standard methods. In order to analyze the correct/incorrect predictions professionally, we have invited the experienced organic chemists to evaluate some random reactions and rewrote the contents of this section on page 14-15 lines 284-315 as follows:

“2.3 Analysis of correct and incorrect predictions

To more comprehensively understand the model performance, we conduct an error analysis of predictions on the USPTO-50k test set. First, 100 random reactions where the results predicted by Graph2Edits differ from the ground-truth reactants are analyzed by professional organic chemists. The assessment gives 85% of the reactions in which the predicted reactants are feasible and considered correct by the chemists, and interestingly, this result is close to the top-1 round-trip accuracy described previously. We here present 30 random examples in Supplementary Table 4 and display that the proposed reactants by Graph2Edits are difficult to distinguish from the ground-truth reactants in terms of reaction feasibility. To further analysis of the incorrect predictions, we then show some reaction samples in Fig. 4 and find that the most common reason for error predictions is ignoring the influence by other functional group in the molecular structure. The prediction by our model in Fig. 4a may fail due to the low reactivity of secondary amine and the steric hindrance of benzyl group. In Fig. 4b, a more nucleophilic aromatic amine group can lead to a completely different product. And also, Graph2Edits sometimes fails to detect multiple reaction sites, possibly resulting in low yield and some by-products (Fig. 4c). These results indicate that there is still significant scope for improvement in the performance of retrosynthesis prediction, such as introducing more chemically meaningful modules to capture the molecular structure information and identify the reactivity of different reaction sites.

In addition, we visualize the top-10 predictions which are different from the ground truth reactants for two cases in Supplementary Fig. 2. We can observe that the common feature of these two products is to have multiple possible reaction centers, and thus can be yielded through a variety of different reaction types. In fact, all top-10 predicted reactants are feasible and can be synthesized by standard methods, although the reaction yields may vary. In Supplementary Fig. 2a, our model provides the options of replacing

'I' with 'Cl' and 'Br' on top-3 and top-7 prediction and amide condensations on top-1 and top-5 prediction. And in Supplementary Fig. 2b, it is worth emphasizing that the ground-truth reactants in USPTO-50k test set is probably wrong, as it is unlikely to introduce stereochemistry far from the reaction center. And Graph2Edits successfully proposes reactions all start from chiral substrates and the top-2 prediction is perfectly fine.”

Fig. 4 Examples of top-1 prediction by Graph2Edits for different errors.

Supplementary Table 4. Top-1 retrosynthesis prediction by Graph2Edits for 30 random target products from USPTO-50k test set on which the prediction is different from the ground truth.

	Product	Ground-truth reactants	Top-1 prediction
1			

2			
3			
4			
5			
6			
7			
8			
9			

10			
11			
12			
13			
14			
15			
16			
17			

18			19			20			21			22			23			24			
25			26			27			28			29			30			

Supplementary Fig. 2. Top-10 predictions by MEGAN for 2 random products from the USPTO-50k test set on which the predictions are different from the ground truth.

Reviewer Point P1.2 — When talking about MEGAN, after rereading the manuscript I cannot see what is the key difference which makes Graph2Edits a better model than MEGAN. As far as I understand, the underlying idea is the same - propose consecutive graph edits to obtain substrates for a given product. MEGAN, however, is very creative - it proposes many "magic" reactions, where some chemically impossible connections of carbon atoms take place. If the authors could show, that their model is better in this aspect (so in fact make better error analysis), that would be an important input.

Reply: We appreciate the reviewer for this valuable comment. In the introduction section, we have described the key differences between our model and MEGAN. On the one hand, although the models generated graph edits by the same auto-regression manner and have certain similarity in the process, in details, our architecture based on D-MPNN greatly simplified the encoder-decoder framework and molecular initial features processing, and the specific algorithm process is provided in Fig. 3b. On the other hand, our model changed the action space from atom level to leaving group level since adding atoms results in a long prediction sequence which increases generation difficulty, and simplified some atomic editing to make it closer to the real chemical reaction. And also, we simplified the network architecture to effectively learn molecular representations and improved the efficiency on applying edits by RDKit for generating the reactants. According to the reviewer's suggestion, we have compared to the baseline model MEGAN and showed a comparison of the reaction examples presented by MEGAN in Supplementary Fig. 3. We have added the related descriptions to the **on page 15 lines 314-320** as follows:

“Furthermore, we conduct a more in-depth performance comparison with the baseline model MEGAN and show a comparison of the reaction examples presented by MEGAN in Supplementary Fig. 3. We observe that the top-1 prediction for the first three reactions by our model are feasible and completely consistent with the ground-truth reactants. And although the top-1 prediction for the last reaction is similar to those by MEGAN, the subsequent top-2 prediction by our method provides a decent alternative.”

Supplementary Fig. 3. Examples of comparison of predictions by MEGAN and Graph2Edits model.

Reviewer Point P1.3 — Also, in the line of comparing to USPTO, the authors make an interesting analysis of similar reactions (section 2.4). However, the similarity measure is based on Tanimoto coefficient. First, I am not sure, how the authors use it on reactions, as the coefficient is calculated for compounds. But more importantly, the Tanimoto coefficient, which is based on some local similarities, may give similar values for chemically distinct compounds (take e.g. heterocyclic compounds), or vice-versa - distinct values for chemically similar structures (where chloride is substituted to iodine, etc.). I doubt if it can efficiently classify reactions. In fact, such classification is usually done based on templates. Did the authors check, that their analysis based on Tanimoto

coefficient indeed separates chemically different reactions? For example, are all electrophilic aromatic substitutions in one class? And electrophilic aromatic substitutions in a separate class? This also applies to the whole section 2.6, where the similarities between reactions are analyzed. Are the clusters really showing different reactions?

Reply: We appreciate the reviewer for pointing the important concern. The reviewer indeed raises an important question about the molecular similarity based on Tanimoto coefficient. In Section 2.4, the Tanimoto similarity is based on the reaction difference fingerprints of the reaction (not for compounds), reported by Schneider et al.⁴ and widely used for reaction classification. In fact, we have misunderstood the **Reviewer 3's** suggestion in the previous revised manuscript on this content, and in this revision, we have conducted the experiments according to the advice given in Reviewer Point P 3.2. The scaffold bias in the original random-split USPTO-50k dataset, where similar molecules appear in both the training and the test set and undergo similar transformations, makes the models achieve high accuracy and does not reflect the true generalization performance of the models. And thus, the USPTO-50k dataset for training and evaluating retrosynthesis prediction models should be split by the Tanimoto similarity of the reaction products and ensured that no reactions in the test set have a product that is within Tanimoto similarity σ of any product from a training set reaction. The results of the re-evaluation on new split USPTO-50k datasets can be seen in Reviewer Point P 3.2, and the contents of this section have been updated accordingly.

Reviewer Point P1.4 — Also, the authors identify the reactions, for which the model proposes more edits with "rare" reactions (line 305). The counterexample is all the heterocycles synthesis - all of them are rather complicated but are really frequent in USPTO (but in USPTO-50k they comprise only 1.8%!).

Reply: We thank the reviewer for this comment. Here, the rare reactions represent a reaction category that account for a very small proportion in our model training dataset

(USPTO-50k) rather than the entire USPTO dataset. Although sometimes these reactions are very common in USPTO or organic reactions, the small amount of these reactions during training gives less opportunities for model learning. In order to avoid ambiguity with rare reactions in organic chemistry, we revised it to ‘**complicated reactions**’

Reviewer Point P1.5 — Also, the discussion of stereochemistry (section 2.4, lines 326+) rises my concerns. After removing the stereochemistry information, "the model has deeply learned the transformation rules of these reactions...". But "these reactions" are just ordinary reactions, for which the stereochemistry information was included. So it would be rather odd if the model could not learn those reactions. Also, with the stereochemistry the "performance drops", but the "top-10 accuracy still achieves 66.2%". Based on the previous discussion of TopK on USPTO, I don't think this argument is convincing. I think, that again, when judging the model efficiency on reactions with stereochemistry, one should check them one by one. The authors finalize the section with "it has an advantage in predicting stereochemical reactions...". There is no such term as "stereochemical reaction". The reaction can be stereospecific, or stereoselective. But I think in this case the authors mean just a reaction, for which the stereo-information is given in USPTO, which is rather random. So I would not draw conclusions here.

Reply: We deeply appreciate the reviewer for this critical comment that indeed improved the quality of this manuscript. In order to address the reviewer's concerns, and due to previous neglect of some reactions with stereochemical change, we have recounted 157 reactions containing the change in stereochemistry in USPTO-50k test set and checked them one by one. Among these reactions, we found that more than half (51.6%) of the reactions gave wrong stereochemical information, which is consistent with the noisy stereochemical data reported by Schwaller et al.¹, and even so, in 82.2% cases, the top-1 prediction proposed by Graph2Edits was considered correct by experienced organic chemists. We show the 30 random reactions in Supplementary

Table 6, and display that our model performed well on the chiral substrate-induced asymmetric reactions (examples 4, 8, 20), chiral auxiliary-induced asymmetric reaction (example 26), asymmetric hydrogenations (examples 24, 30). And we have rewrote the contents of this paragraph on page 17-18 lines 356-369 as follows:

“Stereochemistry plays a significant role in organic chemistry and is also important in drug discovery. It is extremely challenging to predict the change of stereochemistry in the reaction. We count 157 reactions containing the change in stereochemistry in USPTO-50k test set and check them one by one. We found that more than half (51.6%) of ground-truth reactions gave wrong stereochemical information, which is consistent with the noisy stereochemical data reported by Schwaller et al.¹, and in 82.2% of the reactions, the top-1 prediction proposed by Graph2Edits was considered correct by experienced organic chemists. We show the 30 random reactions in Supplementary Table 6, and display that our method performed well on the chiral substrate-induced asymmetric reactions (examples 4, 8, 20), chiral auxiliary-induced asymmetric reaction (example 26), asymmetric hydrogenations (examples 24, 30). Although this stereochemical data set is too limited to claim the performance on stereochemistry, these assessments offer strong evidence that our model has an advantage in predicting stereoselective reactions and can learn some rules of stereochemistry changes.”

Supplementary Table 6. Top-1 retrosynthesis prediction by Graph2Edits for 30 random reactions with stereochemistry change from USPTO-50k test set.

	Product	Ground-truth reactants	Top-1 prediction
1			2			
3			
4			
5			
6			
7			
8			
9			
10			

11			
12			
13			
14			
15			
16			
17			

18			19			20			21			22			23			24			
25			
26			
27			
28			
29			
30			

Reviewer Point P1.6 — There are also some less critical statements I am not sure about. Line 89 - "Transformers have a lower [compared to template-based methods]

computational cost". I am not sure if that is true - transformers are known to be efficient, but rather slow in production, don't they?

Reply: We thank the reviewer for this comment. The sentence "Compared with template-based approaches, template-free methods based on Transformer model are more generalizable and have a lower computational cost." in the manuscript is intended to illustrate that the template-based methods need to extract the reaction templates through preprocessing and match the target molecule with a large number of reaction templates, and this subgraph isomorphism requires a high computational cost^{5,6}. In contrast, the template-free methods bypass templates by learning a direct mapping from the SMILES representations and generate reactant SMILES character by character, and thus have greater generalization potential and a relatively low computational cost. From the model point of view, Transformer is indeed relatively slow in the generation process. To avoid ambiguity, we have revised the sentence on page 5 lines 88-91 as follows:

"Compared with the template-based approaches, the template-free methods directly generate the reactant SMILES character-by-character without subgraph matching computation, which have greater generalization potential and a relatively low computational cost."

Reviewer Point P1.7 — Also, I think that one of the substrates in Fig. 6c should not be aromatic, and the reaction in Fig 7c is in fact dehydration.

Reply: Thanks for your careful checks. We are really sorry for our careless mistakes. In Fig.6c, one of the substrates is indeed cyclohexane, and in Fig. 7c, we have corrected the "Elimination reaction" to "Dehydration reaction". The revised Fig.6 is attached below for review.

Fig. 6 Retrosynthesis reasoning predictions by our model. a Suzuki coupling reaction, **b** Paal-Knorr reaction and **c** Mitsunobu reaction. The ‘T’ represents a termination symbol.

Reviewer Point P1.8 — Finally, I found some typos and grammar problems:

- line 158 - adding A new atom
- line 162 - Rather THAN only
- line 171 - AND is unnecessary
- line 176 - and have been -> WHICH have been
- line 181 - "same" is not needed
- line 279 - leaving group IS
- line 286 - of THE model

- line 357 - Evaluation OF the
- line 456 - unnecessary "in three examples"
- line 457 - pathways AS those
- line 515 - missing whitespace after "where"
- line 544 - unnecessary "the"
- caption to Fig. 5 - there should be violet instead of blue

Reply: We sincerely appreciate the reviewer for careful reading. In our resubmitted manuscript, the typos and grammar problems have been revised and marked in red fonts. Thanks for your correction.

Reviewer Point P1.9 — To sum up, I like the idea of the paper, and I like the analysis and the examples, but I think right now the authors are going too far with their conclusions, which stems from a bit careless usage of tools and metrics in my opinion.

Reply: We thank the reviewer for the positive comments and valuable suggestions. In this revision, we have followed the suggestions made by the reviewer and updated the manuscript accordingly, and hope the reviewer find the revision acceptable.

Reviewer 2

I have one further comment on the accuracy of the model as described in the paper. At the second point on the main contribution, the authors state that D-MPNN can improve the predictive performance. If one sees Table 1, however, the prediction accuracy is higher at top-1 and top-3 (marginally) only but lower at top-5, 10, 50 compared to MPNN-based model. Since retrosynthesis is potentially a one-to-many prediction, higher accuracy at high k can be quite important. Also, can the authors provide some reasoning why higher k accuracy is lower, and only top-1/3 accuracy is improved?

Reply: We thank the reviewer for the valuable comments. Although for the one-to-many retrosynthetic prediction, the accuracy at high k is important as it reflects the effectiveness of the model in searching the space of plausible reactions, the accuracy at lower k indicates that the model tends to find the correct answer with the high probability, and has a higher practicality for assessing the retrosynthetic model performance. Furthermore, when we evaluate the round-trip accuracy, which reflects the fact that a given chemical product could be synthesized by multiple precursors' combinations (see Reviewer Point P 3.1), the accuracies at high k in DMPNN-based model are comparable to the MPNN-based model. Theoretically, the reason for the higher accuracy at higher k in MPNN-based model than DMPNN-based may be that the message passing in MPNN is along the nodes and is undirected, which lead to the redundancy in node messages passing^{7,8} as described in Section 4.2, and the redundancy on the node features may help to improve the probability of predicting the correct leaving group on the reaction centers. To clearly describe this point, we have added a few discussions to the Section 2.2 on page 13 lines 256-261 as follows:

“In addition, although the higher accuracies at higher k have been achieved in MPNN-based models as the redundancy in node messages^{7,8} passing may help to improve the probability of predicting the ground-truth leaving group on the reaction centers, using D-MPNN encoder has a clear advantage over conventional MPNN, yielding

improvements of 1.4 and 2.4 points on top-1 accuracy with and without giving reaction class, respectively.”

Reviewer 3

In general I am satisfied with the modifications to the manuscript and with the answers the authors provided. Two things I would like to point out:

Reply: We thank the reviewer for his/her positive comments. In this revision, we have followed the suggestions made by the reviewer and updated the manuscript accordingly.

Reviewer Point P3.1 — MaxFrag is not much more relevant than topk. I would encourage the authors to use Round-Trip accuracy. While not perfect, can be easily evaluated by training a Transformer model for reaction prediction on USPTO-50k

Reply: We thank the reviewer for this valuable suggestion. We do agree with the reviewer that the round-trip accuracy is a crucial evaluation as it can quantify the percentage of the valid retrosynthetic suggestions. To more comprehensively evaluate our model, we have calculated the round-trip accuracy and compared to the existing state-of-the-art semi-template-based methods, the results were shown in Table 2. From the results we can see that the top-1 round-trip accuracy of our Graph2Edits is comparable to GraphRetro and outperforms MEGAN by a large margin, and Graph2Edits also beats prior semi-template-based models on top-3, -5, -10 and -50 predictions. We have added the related descriptions to the Section 2.2 on page 11 lines 222-229 and page13 lines 269-273 as follows:

“We additionally adopt the round-trip¹ and MaxFrag⁹ accuracy to evaluate the performance of our model. The round-trip accuracy is calculated by comparing the ground-truth product with the product predicted by a forward reaction prediction model using the predicted reactants, and is to evaluate the correctness of the predictions generated by the retrosynthetic model as there might be multiple different reactants can

be used to synthesize the same product. We here use the pretrained forward-synthesis prediction model Molecular Transformer (MT)¹⁰ to evaluate the round-trip accuracy.”

“The results of round-trip and MaxFrag accuracy of our model tested on USPTO-50k are shown in Table 2. The top-1 round-trip accuracy of our model reaches nearly 86%, which is comparable to GraphRetro and outperforms MEGAN by a large margin. And Graph2Edits also beats prior semi-template-based models on top-3, -5, -10 and -50 predictions.”

Table 2 Top-*k* round-trip and MaxFrag accuracy of the proposed Graph2Edits and baselines on USPTO-50k dataset.

Category	Model	Top- k (%)				
		k = 1	3	5	10	50
Round-trip accuracy	MEGAN ^a	82.0	89.9	91.7	94.0	96.4
	GraphRetro ^a	86.0	89.9	90.7	91.4	91.6
	Graph2Edits (MPNN)	84.9	93.6	95.7	96.9	98.9
	Graph2Edits (D-MPNN)	85.9	93.5	95.1	96.4	97.3
MaxFrag accuracy	Aug.Transformer	58.5	73.0	85.4	90.0	-
	LocalRetro	57.8	82.1	89.7	95.0	98.4
	MEGAN	54.2	75.7	83.1	89.2	95.1
	Graph2Edits (MPNN)	56.8	80.3	87.5	92.8	95.4
	Graph2Edits (D-MPNN)	59.2	80.1	86.1	91.3	93.1

^a Denotes that the result is implemented by the open-source code.

Reviewer Point P3.2 — The analysis with Tanimoto similarity is not exactly what I intended to see, because the training set used to train the model seems to be the same from the random splitting. The idea was to generate a new train/validation/test split maximising the diversity between the sets. Similar to this paper: <https://www.nature.com/articles/s41467-021-21895-w>

Reply: We deeply appreciate the reviewer for pointing the important points and sorry for misunderstanding the suggestion at first. In this revised manuscript, we have trained and evaluated the performance of the model on different train/validation/test splits based on Tanimoto similarity and compared with the state-of-the-art semi-template-based models (MEGAN, GraphRetro), the results were shown in Table 4. To split the USPTO-50k dataset via the Tanimoto similarities of the reaction products, the initial dataset was randomly split 85%:15%, for the Tanimoto similarity threshold $\sigma = 0.6$ and $\sigma = 0.4$, the ratios after Tanimoto splitting are 88.3%:11.7% and 95%:5%, respectively. From the Table 4 we can see that the performance of both our Graph2Edits and baselines decrease upon the new train/validation/test split dataset, but our model still outperform MEGAN and GraphRetro by a large margin. We have added the related descriptions to the Section 2.4 on page 16-17 lines 340-355 as follows:

“As revealed by MTEExplainer¹¹, scaffold bias in the USPTO dataset, where similar molecules appear in both the training and the test set and undergo similar transformations, makes the models achieve high accuracy and does not reflect the true generalization performance of the models. To remove the structural bias and further investigate the performance on diverse reaction products, we re-split the USPTO-50k dataset via the Tanimoto similarities¹² of the reaction products to train the retrosynthetic prediction models. Following the Tanimoto-based splitting given by MTEExplainer, the initial USPTO-50k dataset is randomly split 85%:15%, and for the Tanimoto similarity threshold $\sigma = 0.6$ and $\sigma = 0.4$, the ratios after Tanimoto splitting are 88.3%:11.7% and 95%:5%, respectively. We then train our Graph2Edits along with the other semi-template-based models (MEGAN and GraphRetro) on these two datasets. Table 4 shows that although the performance of both our Graph2Edits and the baselines decrease upon the new train/validation/test split datasets, our model still outperform MEGAN and GraphRetro by a large margin. These results show that our model could also achieve relatively good generalization performance on the structurally diverse test set.”

Table 4 Evaluation of single-step retrosynthetic models on different train-test splits of USPTO-50k dataset.

Data split	Model	Top- k accuracy (%)			
		k = 1	3	5	10
Original random split	MEGAN	48.1	70.7	78.4	86.1
	GraphRetro	53.7	68.3	72.2	75.5
	Graph2Edits (D-MPNN)	55.1	77.3	83.4	89.4
Tanimoto similarity < 0.6	MEGAN ^a	47.0	69.2	76.2	83.6
	GraphRetro ^a	49.1	63.2	66.9	69.1
	Graph2Edits (D-MPNN)	52.0	75.6	83.2	89.4
Tanimoto similarity < 0.4	MEGAN ^a	45.4	68.4	76.9	84.6
	GraphRetro ^a	44.2	56.2	58.7	59.6
	Graph2Edits (D-MPNN)	47.5	71.7	80.1	88.0

^a Denotes that the result is implemented by the open-source code with well-tuned hyperparameters.

References

- 1 Schwaller, P. et al. Predicting retrosynthetic pathways using transformer-based models and a hyper-graph exploration strategy. *Chemical science* **11**, 3316-3325 (2020).
- 2 Chen, B., Li, C., Dai, H. & Song, L. Retro*: learning retrosynthetic planning with neural guided A* search. In: *International Conference on Machine Learning*. 1608-1616 (2020).
- 3 Toniato, A., Schwaller, P., Cardinale, A., Geluykens, J. & Laino, T. Unassisted noise reduction of chemical reaction datasets. *Nature Machine Intelligence* **3**, 485-494 (2021).

- 4 Schneider, N., Lowe, D. M., Sayle, R. A. & Landrum, G. A. Development of a novel fingerprint for chemical reactions and its application to large-scale reaction classification and similarity. *Journal of chemical information and modeling* **55**, 39-53 (2015).
- 5 Dong, J., Zhao, M., Liu, Y., Su, Y. & Zeng, X. Deep learning in retrosynthesis planning: datasets, models and tools. *Brief Bioinform* **23**, bbab391 (2022).
- 6 Shi, C., Xu, M., Guo, H., Zhang, M. & Tang, J. A graph to graphs framework for retrosynthesis prediction. In: *International conference on machine learning*. 8818-8827 (2020).
- 7 Yang, K. et al. Analyzing learned molecular representations for property prediction. *Journal of chemical information and modeling* **59**, 3370-3388 (2019).
- 8 Nyamabo, A. K., Yu, H., Liu, Z. & Shi, J.-Y. Drug–drug interaction prediction with learnable size-adaptive molecular substructures. *Brief Bioinform* **23**, bbab441 (2022).
- 9 Tetko, I. V., Karpov, P., Van Deursen, R. & Godin, G. State-of-the-art augmented NLP transformer models for direct and single-step retrosynthesis. *Nature communications* **11**, 1-11 (2020).
- 10 Schwaller, P. et al. Molecular transformer: a model for uncertainty-calibrated chemical reaction prediction. *ACS central science* **5**, 1572-1583 (2019).
- 11 Kovács, D. P., McCorkindale, W. & Lee, A. A. Quantitative interpretation explains machine learning models for chemical reaction prediction and uncovers bias. *Nature Communications* **12**, 1695 (2021).
- 12 Bajusz, D., Rácz, A. & Héberger, K. Why is Tanimoto index an appropriate choice for fingerprint-based similarity calculations? *Journal of cheminformatics* **7**, 1-13 (2015).

REVIEWER COMMENTS

Reviewer #1 (Remarks to the Author):

I deeply appreciate the effort the authors put into the manuscript. All my doubts are solved right now. I have only a few last remarks I've spotted after reading the manuscript again, which authors may want to introduce:

- Line 180 - I believe there should be a comma between "and" in "reactions for the training, validation and test sets"
- There is additional space in "Edit Atoms" list in Fig 1c.
- It would be good to specify the errors in the caption of Fig. 4 (what is a, b, and c)
- Fig 10, caption - I believe there should be Oxford Comma before "and".

I hope, that although my remarks forced authors to huge additional amount of work, they find it valuable, and the effort will eventually pay off as the paper can be important for the community.

Reviewer #2 (Remarks to the Author):

In this revision, it is good that the authors added the round-trip accuracy (that can partly address many possible answers to the retrosynthesis problem for practicality). However, one question is, in Table 2, why did the authors not include the results for LocalRetro? According to the LocalRetro paper, the LocalRetro's round-trip accuracy for USPTO-50K seems much higher than other methods compared here (including Graph2Edits). If no apparent reasons, these should be added and discussed.

In line 67~73, the authors mention that, "Recently, LocalRetro evaluated the suitable local templates (atom/bond templates) at all enumerated possible reaction centers of a target molecule ... cannot be extended to large scale template sets because of the expensive computational cost". If I understood the paper correctly, Localretro does not enumerate local templates to all possible reaction centers, but only to the predicted reaction centers to save computational time.

REVIEWER COMMENTS

Reviewer #1 (Remarks to the Author):

I deeply appreciate the effort the authors put into the manuscript. All my doubts are solved right now. I have only a few last remarks I've spotted after reading the manuscript again, which authors may want to introduce:

- Line 180 - I believe there should be a comma between "and" in "reactions for the training, validation and test sets"
- There is additional space in "Edit Atoms" list in Fig 1c.
- It would be good to specify the errors in the caption of Fig. 4 (what is a, b, and c)
- Fig 10, caption - I believe there should be Oxford Comma before "and".

I hope, that although my remarks forced authors to huge additional amount of work, they find it valuable, and the effort will eventually pay off as the paper can be important for the community.

Reviewer #2 (Remarks to the Author):

In this revision, it is good that the authors added the round-trip accuracy (that can partly address many possible answers to the retrosynthesis problem for practicality). However, one question is, in Table 2, why did the authors not include the results for LocalRetro? According to the LocalRetro paper, the LocalRetro's round-trip accuracy for USPTO-50K seems much higher than other methods compared here (including Graph2Edits). If no apparent reasons, these should be added and discussed.

In line 67~73, the authors mention that, "Recently, LocalRetro evaluated the suitable local templates (atom/bond templates) at all enumerated possible reaction centers of a target molecule ... cannot be extended to large scale template sets because of the expensive computational cost". If I understood the paper correctly, Localretro does not

enumerate local templates to all possible reaction centers, but only to the predicted reaction centers to save computational time.

Reviewer 1

Reviewer Point P1.1 — I deeply appreciate the effort the authors put into the manuscript. All my doubts are solved right now. I have only a few last remarks I've spotted after reading the manuscript again, which authors may want to introduce:

- Line 180 - I believe there should be a comma between "and" in "reactions for the training, validation and test sets"
- There is additional space in "Edit Atoms" list in Fig 1c.
- It would be good to specify the errors in the caption of Fig. 4 (what is a, b, and c)
- Fig 10, caption - I believe there should be Oxford Comma before "and".

Reply: We thank the reviewer for the careful reading again and for pointing out the grammatical errors that indeed improved the quality of this manuscript. In our resubmitted manuscript, the grammar problems have been revised and marked in red fonts. Thanks for your correction.

- The sentence on page 13 line 180 was revised as:

“divide it into 40k, 5k, 5k reactions for the training, validation, and test sets, respectively.”

- The caption of Fig. 4 was revised on page 46 lines 878-880 as:

“**Fig. 4 Examples of top-1 prediction by Graph2Edits for different errors. a** The low reactivity of secondary amine and the steric hindrance of benzyl group, **b** Ignoring a more nucleophilic aromatic amine group, and **c** Fail to detect multiple reaction sites.”

- The caption of Fig. 10 was revised on page 52 lines 914-915 as follows:

“**Fig. 10 Multistep retrosynthesis predictions by Graph2Edits. a** Nirmatrelvir, **b** Osimertinib, and **c** Lenalidomide.”

- The revised Fig 1 was attached below for review:

a. Atom center reaction

b. Bond center reaction

c. Multiple centers reaction

Fig. 1 The examples of derived edits sequence in retro-reactions. **a** Atom center reaction, **b** Bond center reaction, and **c** Multiple centers reaction.

Reviewer Point P1.2 — I hope, that although my remarks forced authors to huge additional amount of work, they find it valuable, and the effort will eventually pay off as the paper can be important for the community.

Reply: We sincerely appreciate the reviewer for the thorough review, constructive comments, and valuable suggestions. It is very helpful to improve the quality of our manuscript, and we have learned a lot from your comments. Thank you again.

Reviewer 3

Reviewer Point P2.1 — In this revision, it is good that the authors added the round-trip accuracy (that can partly address many possible answers to the retrosynthesis problem for practicality). However, one question is, in Table 2, why did the authors not include the results for LocalRetro? According to the LocalRetro paper, the LocalRetro's round-trip accuracy for USPTO-50K seems much higher than other methods compared here (including Graph2Edits). If no apparent reasons, these should be added and discussed.

Reply: We thank the reviewer for this valuable comments. In fact, we have noticed this problem before. Perhaps due to the different details of the calculation methods, the round-trip accuracies of the LocalRetro model for USPTO-50k seem to be higher than our results. And in the code link (<https://github.com/kaist-amsg/LocalRetro>) provided in the LocalRetro paper, the author did not provide their code for calculating the round-trip accuracy, nor did they provide an available trained LocalRetro model, so we cannot reproduce their calculation results and compare them fairly with our model. To address this concern, we have now uploaded the process code for calculating the round-trip accuracy in our model to GitHub (<https://github.com/Jamson-Zhong/Graph2Edits>), and have added the round-trip accuracy results of LocalRetro model to Table 2 as a reference. The related illustration was attached to the Section 2.2 on page 13 lines 272-277 as follows:

“Additionally, perhaps due to the detailed difference of the calculation methods, the round-trip accuracies of the LocalRetro for USPTO-50k seem to be higher than our results. As there is no related code for calculating the round-trip accuracy in LocalRetro GitHub, in order to make a fair comparison in the semi-template-based methods, we calculate the round-trip accuracies based on the trained models provided by MEGAN and GraphRetro, and provide the LocalRetro's round-trip accuracy results as a reference.”

Table 2 Top-*k* Round-Trip and MaxFrag accuracy of the proposed Graph2Edits and baselines on USPTO-50k dataset.

Category	Model	Top- k (%)				
		k = 1	3	5	10	50
Round-Trip accuracy	LocalRetro ^a	89.5	97.9	99.2	-	-
	MEGAN ^b	82.0	89.9	91.7	94.0	96.4
	GraphRetro ^b	86.0	89.9	90.7	91.4	91.6
	Graph2Edits (MPNN)	84.9	93.6	95.7	96.9	98.9
	Graph2Edits (D-MPNN)	85.9	93.5	95.1	96.4	97.3
MaxFrag accuracy	Aug. Transformer	58.5	73.0	85.4	90.0	-
	LocalRetro	57.8	82.1	89.7	95.0	98.4
	MEGAN	54.2	75.7	83.1	89.2	95.1
	Graph2Edits (MPNN)	56.8	80.3	87.5	92.8	95.4
	Graph2Edits (D-MPNN)	59.2	80.1	86.1	91.3	93.1

^aThe results are taken from the LocalRetro paper. ^b The results are implemented based on the available trained models in the open-source code.

Reviewer Point P2.2 — In line 67~73, the authors mention that, "Recently, LocalRetro evaluated the suitable local templates (atom/bond templates) at all enumerated possible reaction centers of a target molecule ... cannot be extended to large scale template sets because of the expensive computational cost". If I understood the paper correctly, Localretro does not enumerate local templates to all possible reaction centers, but only to the predicted reaction centers to save computational time.

Reply: We deeply appreciate the reviewer for pointing the important points. In the caption of Figure 2 of the LocalRetro paper, the author wrote "Finally, the score of each local reaction template on each atom and bond is predicted by the atom reaction template classifier and bond reaction template classifier. The predicted reactants are

obtained by applying the predicted local reaction template on the predicted atoms and bonds and ranked by their predicted scores.”, and indicates that the LocalRetro model has all atom template or bond template related prediction scores for each atom or bond in the product molecule, and then ranks the reaction centers according to the scores. Additionally, the number of local reaction templates in the USPTO-50k training set with 40K reactions is 731, while the number in the USPTO-MIT training set with 400K reactions is 21081. As the dataset grows, the number of local reaction templates will increase significantly, thus increasing the computational cost. We here revised the related descriptions on page 4 lines 67-70 as follows:

“Recently, LocalRetro evaluated the suitable local templates (atom/bond templates) at the predicted reaction centers of a target molecule and considered the nonlocal effects of chemical reactions using global reactivity attention, which achieved the state of the art in the template-based methods.”

REVIEWERS' COMMENTS

Reviewer #2 (Remarks to the Author):

The authors have addressed my previous comments, and I have no further comments.

REVIEWER COMMENTS

Reviewer #2 (Remarks to the Author):

The authors have addressed my previous comments, and I have no further comments.

Reviewer 2

Reviewer Point — The authors have addressed my previous comments, and I have no further comments.

Reply: We sincerely appreciate the reviewer for the thorough review, constructive comments, and valuable suggestions. It is very helpful to improve the quality of our manuscript, and we have learned a lot from your comments. Thank you again.